# Isolation of small extracellular vesicles from small volumes of blood plasma using size exclusion chromatography and density gradient ultracentrifugation

**Fang Kong[1]\*, Megha Upadya[1], Andrew SW Wong[2], Rinkoo Dalan[3], Ming Dao[1,4]\***

[1]School of Biological Sciences, Nanyang Technological University, Singapore, Singapore; [2]Facility for Analysis, Characterisation, Testing and Simulation, Nanyang Technological University, Singapore, Singapore; [3]Lee Kong Chian School of Medicine, Nanyang Technological University, Singapore, Singapore; [4]Department of Material Science and Engineering, Massachusetts Institute of Technology, Cambridge, United States

## eLife Assessment

This work provides a simple, rapid and **valuable** protocol for the isolation of small extracellular vesicles from small volumes of plasma, using two well-known methodologies, in tandem: size exclusion chromatography (SEC) and density gradient ultracentrifugation (DGUC). The authors exhaustively test these methodologies separately and in combination, showing superior results for the SEC-DGUC in terms of purity and yield. The results obtained in this work are **convincing**, using multiple state-of-art methodologies for the characterization of the isolates that support their conclusions.

**\*For correspondence:** kongfang@gmail.com (FK); mingdao@mit.edu (MD)

**Competing interest:** The authors declare that no competing interests exist.

**Sent for Review** 30 October 2023
**Preprint posted** 01 November 2023
**Reviewed preprint posted** 16 January 2024
**Reviewed preprint revised** 08 April 2025
**Version of Record published** 22 May 2026

**Abstract** Small extracellular vesicles (sEVs) are heterogeneous biological vesicles released by cells under both physiological and pathological conditions. Due to their potential as valuable diagnostic and prognostic biomarkers in human blood, there is a pressing need to develop effective methods for isolating high-purity sEVs from the complex milieu of blood plasma, which contains abundant plasma proteins and lipoproteins. Size exclusion chromatography (SEC) and density gradient ultracentrifugation (DGUC) are two commonly employed isolation techniques that have shown promise in addressing this challenge. In this study, we aimed to determine the optimal combination and sequence of SEC and DGUC for isolating sEVs from small plasma volumes, in order to enhance both the efficiency and purity of the resulting isolates. To achieve this, we compared sEV isolation using two combinations: SEC-DGUC and DGUC-SEC, from unit volumes of 500 µL plasma. Both protocols successfully isolated high-purity sEVs; however, the SEC-DGUC combination yielded higher sEV protein and RNA content. We further characterized the isolated sEVs obtained from the SEC-DGUC protocol using flow cytometry and mass spectrometry to assess their quality and purity. In conclusion, the optimized SEC-DGUC protocol is efficient, highly reproducible, and well suited for isolating high-purity sEVs from small blood volumes.

## Introduction

In recent years, extracellular vesicles (EVs) have garnered significant scientific attention due to their potential as biomarkers and applications in targeted drug delivery (*EL Andaloussi et al., 2013*). These

vesicles are secreted by all cell types and have been detected in various types of bodily fluids such as blood, urine, saliva, and feces (*Witwer et al., 2013*). Comprising a phospholipid bilayer with a composition similar to the cell of origin, EVs carry a diverse range of molecules, including proteins, nucleic acids, and lipids (*Colombo et al., 2014*). They play crucial roles in both normal physiological processes and pathological conditions.

EVs can be classified into three main subgroups based on their biogenesis: exosomes, microvesicles (MVs), and apoptotic bodies. Due to the challenges in distinguishing between exosomes and MVs smaller than 150 nm, which share similar size, density, and protein markers, they are collectively referred to as small extracellular vesicles (sEVs) (*Witwer and Théry, 2019*). These sEVs are of particular interest in the field of biomarker research and targeted drug delivery, given their ubiquitous presence and functional relevance in various biological processes.

Blood plasma sEVs provide valuable information for diagnosis, prognosis, and homeostasis (*EL Andaloussi et al., 2013*). However, isolation of sEVs from plasma is hindered by two major constituents: plasma proteins and lipoproteins (*Simonsen, 2017*; *Muller et al., 2014*; *van der Pol et al., 2018*). Contamination of these constituents in sEV isolates can affect the understanding of the role of sEVs as carriers of genetic information and protein antigens in human physiology and pathology. Numerous techniques, including differential ultracentrifugation (dUC), size exclusion chromatography (SEC), ultrafiltration, immunoaffinity isolation, and microfluidics (*Wu et al., 2017*), are used to isolate sEVs from plasma. SEC (*Böing et al., 2014*) and density gradient ultracentrifugation (DGUC) are particularly popular due to their unique advantages in isolating sEVs based on size and buoyant density, respectively. In many situations involving human blood, such as routine blood tests, only a small volume of blood (1–10 mL) is collected, which yields a limited amount of plasma (500 μL to 5 mL). To obtain high-purity sEVs with a reasonable yield from this small volume of plasma, it is essential to establish a repeatable and reliable isolation protocol. Additionally, the isolation method should be simple, practical, and involve commonly available equipment to ensure ease of use.

To address this challenge, we explored the combination of SEC and DGUC, which has been shown to lead to high-purity sEV isolates by effectively removing both lipoproteins and plasma proteins (*Théry et al., 2006*; *Zhang et al., 2020*; *Vergauwen et al., 2021*; *Karimi et al., 2018*; *Wei et al., 2020*). However, the ideal sequence for these methods in isolating sEVs from small volumes of plasma remains unclear. In this study, we compared the sequences of SEC-DGUC and DGUC-SEC to determine the better option in terms of purity and yield. Our study reinforces previous findings (*Vergauwen et al., 2021*; *Driedonks et al., 2020*) on the efficacy of SEC in eliminating plasma proteins and high-density lipoproteins (HDLs), as well as the utility of DGUC in distinguishing sEVs from lipoproteins.

We examined the performance of two protocols, SEC-DGUC and DGUC-SEC, in terms of sEV purity and yield. Both methods produced high-purity sEVs, but the SEC-DGUC protocol outperformed DGUC-SEC in terms of sEV protein and RNA yield.

In order to enhance the efficiency of the DGUC process, we optimized it using a tailored density gradient in a 1.5 mL tube format and a fixed-angle rotor. This adjustment significantly reduced the ultracentrifugation time to just 2 hr. Our optimized approach enabled the harvesting of sEVs across an extensive density range (>1.08 g/mL), achieving high-purity isolation in a short span of 3 hr. This study thus offers an efficient protocol for sEV isolation that combines optimal yield and high purity. Subsequently, we further characterized the isolated sEVs using flow cytometry (FCM) and mass spectrometry techniques. These additional investigations provided further confirmation of the sEVs' purity and structural integrity.

In conclusion, our study highlights the superiority of the SEC-DGUC protocol over DGUC-SEC in terms of purity and yield when isolating sEVs from as little as 500 μL of plasma. This optimized protocol is well suited for clinical applications requiring high-purity sEVs from small blood volumes and is easily adaptable for various research and clinical settings due to its simplicity, practicality, and use of commonly available equipment.

## Results

### SEC isolated particles from plasma with 1% sEV content

We first assessed the efficacy of SEC in isolating particles, particularly sEVs, from plasma (*Figure 1A*). A volume of 500 μL plasma was loaded onto a SEC column with a maximum loading capacity of 2 mL.

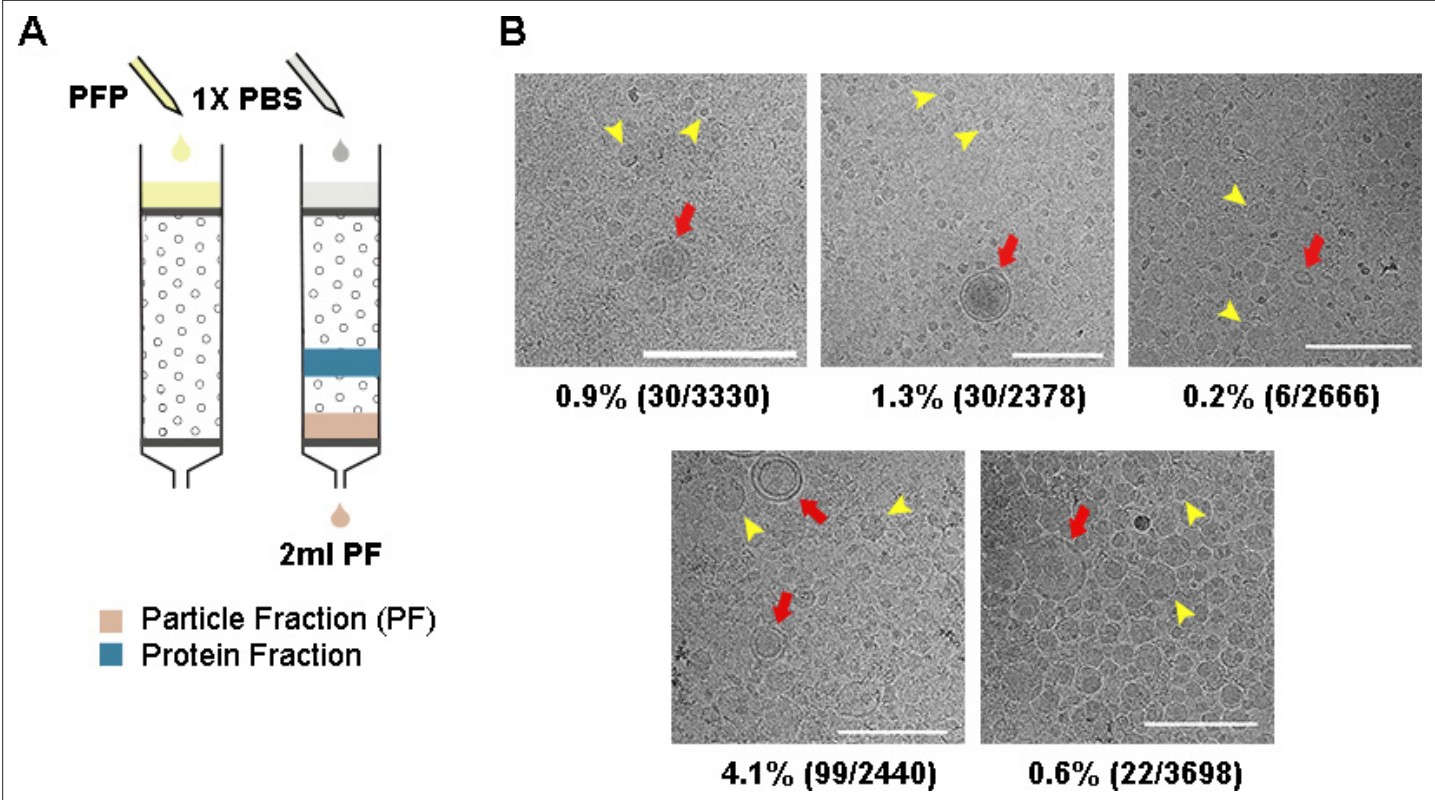

**Figure 1.** Size exclusion chromatography (SEC) was effective in removing plasma proteins and high-density lipoprotein (HDL) but not low-density lipoproteins. (**A**) 500 μL of plasma was loaded on a SEC column and particle fractions (PFs) were collected. (**B**) Cryo-EM images of PFs obtained from the plasma of five healthy individuals. Vesicles could be easily identified, and the images revealed a clean background suggesting minimum protein contamination. Moreover, HDL was sparse in the cryo-EM images, indicating that they were also largely removed. Furthermore, small extracellular vesicles (sEVs), having clearly defined bilayers (red arrows), were easily distinguishable from lipoproteins (representative particles marked by yellow arrows). The ratio of counted sEVs to all vesicles is shown under each image. All scale bars represent 200 nm.

The online version of this article includes the following source data and figure supplement(s) for figure 1:

**Figure supplement 1.** Size exclusion chromatography (SEC) elution profiles, EM images of the particle fraction (PF), and particle size distributions.

**Figure supplement 1—source data 1.** Numerical data corresponding to size exclusion chromatography (SEC) elution profiles, nanoparticle tracking analysis (NTA) size distributions, and transmission electron microscopy (TEM)-derived particle size histograms.

We collected and pooled four fractions, fractions 7–10 (2 mL total volume), as the 'particle fraction' (PF) for each SEC run. This procedure is consistent with the particle concentration profiles (*Figure 1—figure supplement 1A*) and follows the SEC column manual's guidelines. Nanoparticle tracking analysis (NTA) measurement of these PFs, obtained from 10 blood plasma samples, demonstrated particle concentrations ranging from $9.6×10^8$ to $5.5×10^{11}$ mL$^{-1}$.

Cryo-EM images of PFs isolated from the plasma of five healthy individuals were used to analyze the content of the PFs (*Figure 1B*). We observed distinct sEVs characterized by bilayer membranes, although they were outnumbered by a large number of lipoproteins (*Figure 1B*, *Figure 1—figure supplement 1C*). The average proportion of sEVs in the PFs was 1%. Notably, HDLs (7–13 nm) (*Zhang et al., 2011*) were observed quite sparingly in the cryo-images of PFs, suggesting a substantially lower presence compared to the other particle populations. On the other hand, the size of the lipoproteins in PF mainly comprised of two populations centered at 20–25 nm and 40–50 nm, corresponding to the sizes of LDL and IDL/VLDL, respectively (*Figure 1—figure supplement 1F*).

Our data demonstrated that SEC effectively removed plasma proteins and HDLs. However, sEVs constituted only 1% of the particles in the PFs, with the remaining 99% consisting of a mixture of LDL and IDL/VLDL lipoproteins.

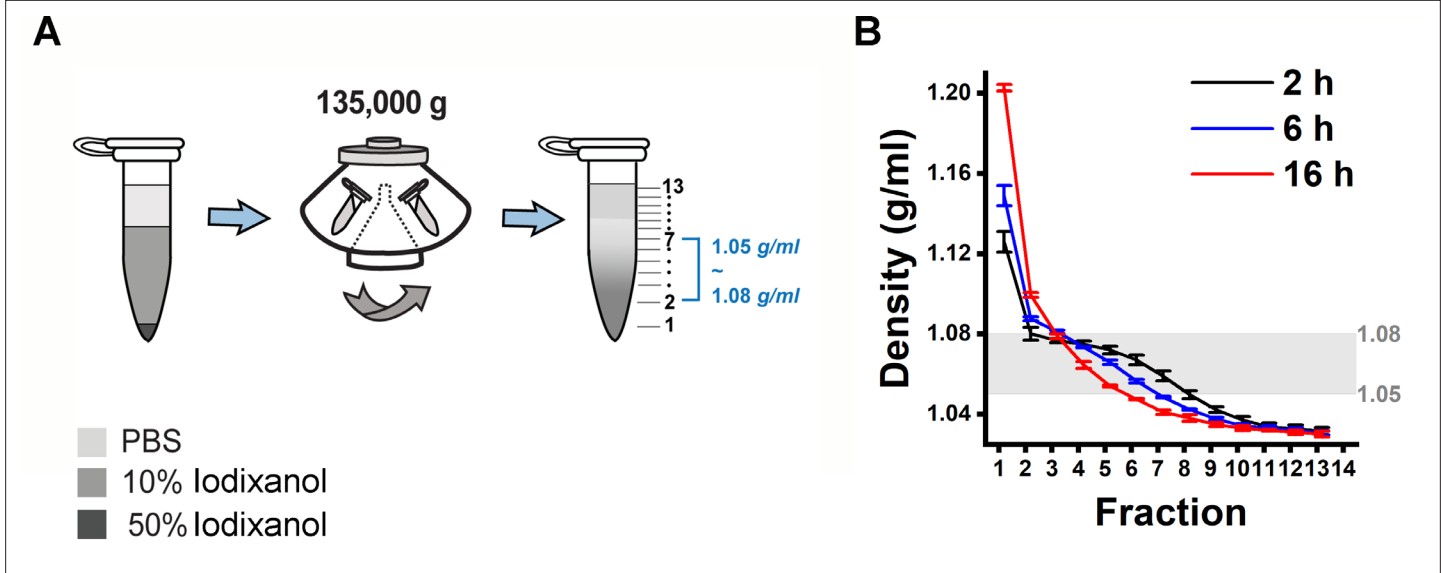

**Figure 2.** Design of a density gradient setup in a small volume format. (**A**) 500 μL phosphate-buffered saline (PBS) was overlaid on 800 μL 10% iodixanol solution and 20 μL 50% iodixanol cushion. The 1.5 mL tubes were subjected to ultracentrifugation using a fixed-angle rotor at an average speed of 135,000×*g* for 2, 6, and 16 hr at 4°C to establish a density gradient profile. (**B**) Density gradient profiles along the 1.5 mL tubes after ultracentrifugation. The 2 hr spinning time gave a density profile with the largest separation zone of 1.05–1.08 g/mL and smallest small extracellular vesicle (sEV) zone of >1.08 g/mL. The data shows the average ± standard deviation of seven repeats for 2 hr, three repeats each for 6 and 16 hr.

The online version of this article includes the following source data and figure supplement(s) for figure 2:

**Source data 1.** Numerical data corresponding to density gradient profiles across fractions under different centrifugation durations.

**Figure supplement 1.** Distribution of small extracellular vesicles (sEVs) and lipoproteins within the density gradient after density gradient ultracentrifugation (DGUC).

**Figure supplement 1—source data 1.** Numerical data corresponding to density values and particle concentration across gradient fractions.

### DGUC density gradient design inside a 1.5 mL Eppendorf tube

Addressing the challenges associated with handling small volumes of plasma in DGUC requires an optimized approach, ideally involving the use of a smaller tube format, such as a 1.5 mL Eppendorf tube. For such optimization, it is critical to accurately identify the density zones of sEVs and lipoproteins, including their overlapping regions, to ensure effective separation.

To fulfill this initial objective, a DGUC experiment was conducted using a swing-bucket in a conventional 12 mL tube. The PF from SEC of 6 mL plasma served as the starting material, ensuring ample substance for subsequent transmission electron microscopy (TEM) analysis (*Figure 2—figure supplement 1*, Materials and methods). After the DGUC, the 12 mL tube was fractionated into 42 distinct fractions, each of which underwent nanoparticle tracking analysis (NTA), TEM, and density measurements. This enabled correlation of the density profile with particle concentrations and TEM images, fostering an understanding of how sEVs separate from lipoproteins within the established density gradient (*Figure 2—figure supplement 1*). The analysis delineated three discrete density zones. The first zone, with densities less than 1.05 g/mL, was characterized by high particle counts and a predominance of lipoproteins, yet devoid of sEVs. The second, a separation zone, spanning from 1.05 g/mL to 1.08 g/mL, displayed a reduction in particle numbers, emergence of sEVs, and a continued dominance of lipoproteins. The final zone, extending beyond 1.08 g/mL and up to 1.2 g/mL, marked as the sEV zone, exhibited a further drop in particle counts, a sharp decrease in lipoproteins, and a marked increase in the presence of sEVs.

Building on the findings from the 12 mL tube format experiment, we aimed to establish a density gradient within a more compact, 1.5 mL tube. This was accomplished by layering an 800 μL 10% iodixanol solution over a 20 μL 50% iodixanol cushion (*Figure 2A*). Subsequently, the assembled density gradient was calibrated with an additional 500 μL of phosphate-buffered saline (PBS) followed by DGUC, utilizing a fixed-angle rotor for durations of 2, 6, and 16 hr (see Materials and methods).

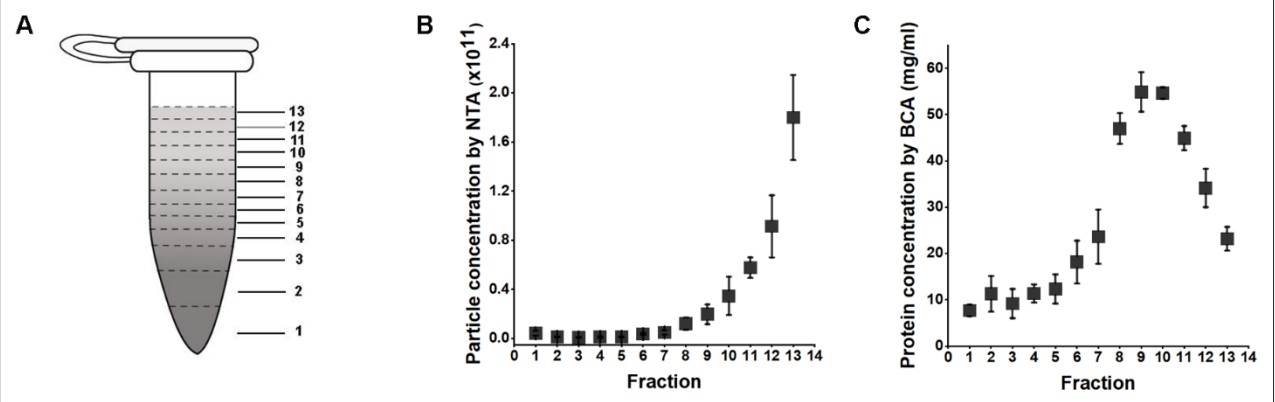

**Figure 3.** Particle and protein concentrations along the 1.5 mL tube after applying density gradient ultracentrifugation (DGUC) to plasma. (**A**) The 1.5 mL tube was fractionated into 13 fractions. (**B**) Particle concentration ± SD measured by nanoparticle tracking analysis (NTA) of different fractions (n=3). (**C**) Protein concentrations ± SD measured by BCA of different fractions (n=3).

The online version of this article includes the following source data for figure 3:

**Source data 1.** Numerical data corresponding to particle concentration (NTA) and protein concentration (BCA) across fractions.

Following the small-tube format DGUC, the tube was fractionated into 13 parts of 100 µL each, with the first fraction at 120 µL to include the 20 µL 50% iodixanol cushion (**Figure 2A**). The density profiles displayed notable consistency across different runs at the tested time intervals of 2, 6, and 16 hr (**Figure 2B**). As the ultracentrifugation duration increased, the profiles evolved from a stepwise to a more continuous pattern. Within the 1.5 mL tube, the potential separation zone (1.05–1.08 g/mL) slimmed down from a region of 700 µL to 500 µL and 300 µL as the ultracentrifugation period extended from 2 to 6 and 16 hr, respectively. Concurrently, the potential sEV zone (>1.08 g/mL) expanded from 120 µL during a 2 hr ultracentrifugation to 220 µL for both 6 and 16 hr. These findings indicated that a 2 hr ultracentrifugation period resulted in the broadest separation zone and the smallest sEV zone.

Given the insights gained from these experiments, it appears that optimizing the DGUC process for a smaller, 1.5 mL tube format and initially setting the ultracentrifugation period at 2 hr could be effective for isolating sEVs.

## DGUC isolated sEVs with significant protein contamination

Building upon these observations, the effectiveness of the 1.5 mL tube format DGUC was evaluated for direct sEV isolation from plasma. A 500 µL plasma sample was subjected to the 2 hr DGUC protocol, with the tube subsequently divided into 13 fractions. Each fraction's particle and protein concentrations were subsequently measured (**Figure 3**). The high-density bottom fraction (>1.08 g/mL), which contained sEVs, was denoted as plasma-DGUC-1. The particle and protein concentration profiles revealed that the plasma-DGUC-1 fraction successfully eliminated the majority of proteins and a significant number of particles (**Figure 3**). However, despite the substantial protein removal, the protein concentration in plasma-DGUC-1 remained relatively high at 7.7 mg/mL. This indicates that while DGUC effectively separates low-density particles from those of high density, a considerable amount of protein still remains within the sEV density zone.

## SEC-DGUC protocol isolated high-purity sEVs

Recognizing the unique advantages and drawbacks of SEC and DGUC, it became clear that a strategic integration of these methods could potentially enhance the isolation of high-purity sEVs from blood plasma. With this perspective, we adopted a combined SEC-DGUC method. Specifically, we subjected PFs, obtained from SEC of a 500 µL plasma sample, to the small-tube format DGUC protocol (**Figure 4A**). The tube contents were systematically divided into 13 fractions, each of which was subjected to NTA, TEM, and density measurements. As expected, fraction 1, comprising the bottom 120 µL, showed a high density (1.10 g/mL) and was populated with high-purity sEVs (**Figure 5**). Fractions 2–7, within the 1.05–1.08 g/mL density zone, displayed sparse sEV presence amidst lipoproteins,

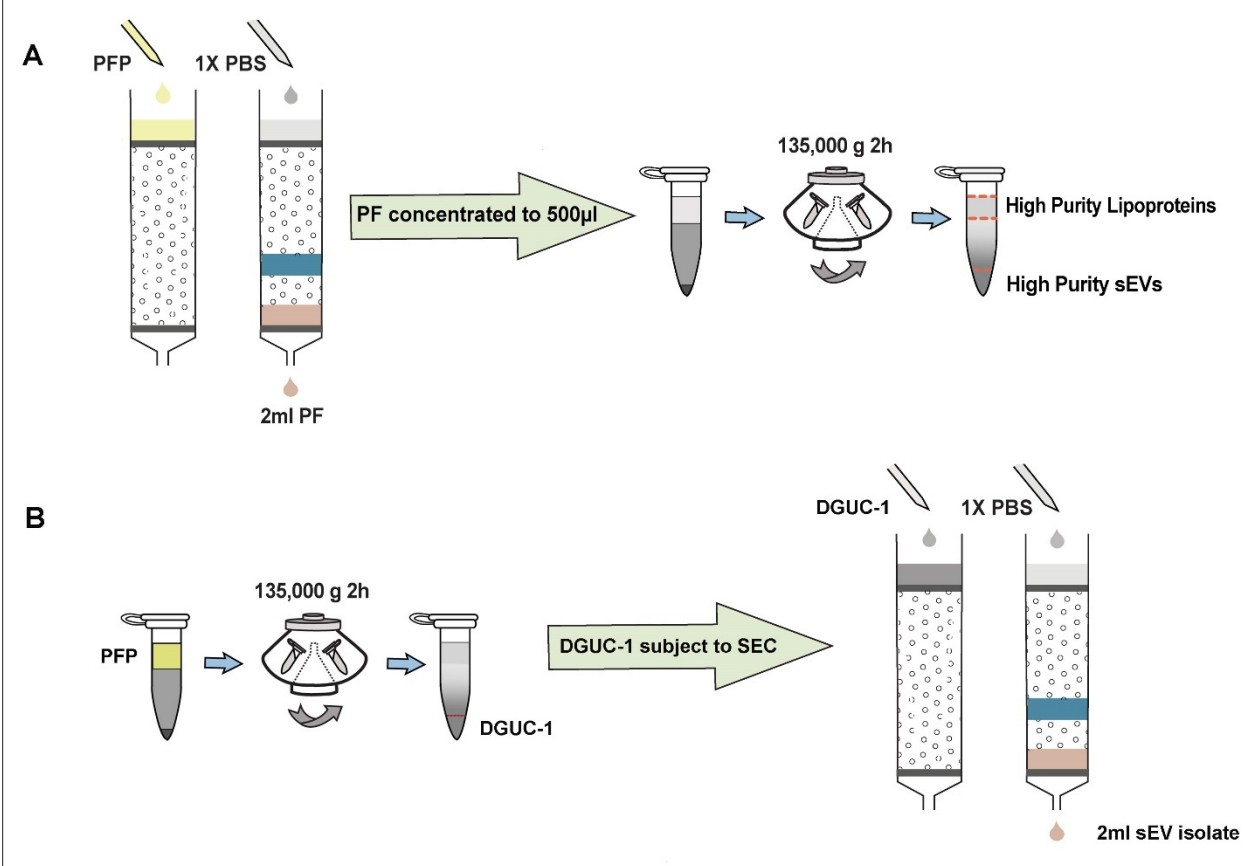

**Figure 4.** Schematic representation of the size exclusion chromatography (SEC)-density gradient ultracentrifugation (DGUC) and DGUC-SEC protocols. (**A**) Illustrates the SEC-DGUC protocol sequence. Starting from the left, SEC is used initially to separate plasma proteins and high-density lipoprotein (HDL). Subsequently, DGUC is employed to separate low-density lipoproteins (i.e. IDL, VLDL, LDL) from small extracellular vesicles (sEVs). (**B**) Depicts the DGUC-SEC protocol, wherein DGUC is initially employed to segregate high-density components from those of lower density, before SEC is applied to partition any remaining proteins from the sEVs.

while the remaining fractions (8–13) were dominated by lipoproteins (*Figure 5*, *Figure 5—figure supplement 1*). Distinct NTA size histograms for fraction 1 further highlighted its divergence from the remaining fractions, which were overwhelmed by lipoproteins (*Figure 5—figure supplement 2B*). As a result, fraction 1 was identified as the final sEV isolate derived from the plasma sample using the SEC-DGUC protocol.

To further substantiate the purity of the sEV isolates obtained through SEC-DGUC protocol, we embarked on measuring the total RNA content of the fractions and conducting western blot analysis (see Materials and methods). For these experiments, we simplified the fractionation of the 1.5 mL tube into four distinct fractions, denoted as SEC-DGUC-1–4 (*Figure 5*, right side, see Materials and methods). Total RNA analysis revealed that SEC-DGUC-1 had a higher concentration of RNA compared to the other three fractions (*Figure 5*). The western blot analysis showed an abundant presence of sEV markers such as CD63, CD81, CD9, TSG101, and Flotillin-1 in SEC-DGUC-1, while the lipoprotein marker, ApoB, was notably low. This contrasted with SEC-DGUC-2–4, where ApoB was abundant, reinforcing the high purity of sEVs in SEC-DGUC-1. Notably, SEC-PF exhibited a high level of ApoB and low expression of sEV markers (*Figure 6B*).

To quantify the purity of sEVs in SEC-DGUC-1, cryo-EM was employed to discern sEVs and lipoproteins (*Figure 6D*, *Figure 6—figure supplement 1*). The purity of sEVs, defined as the percentage of sEVs among all particles, was calculated to be 40%, where the other 60% of particles were mainly small lipoproteins. Considering that SEC-PFs were comprised of only ~1% of sEVs, the DGUC managed to eliminate over 98% of the lipoproteins.

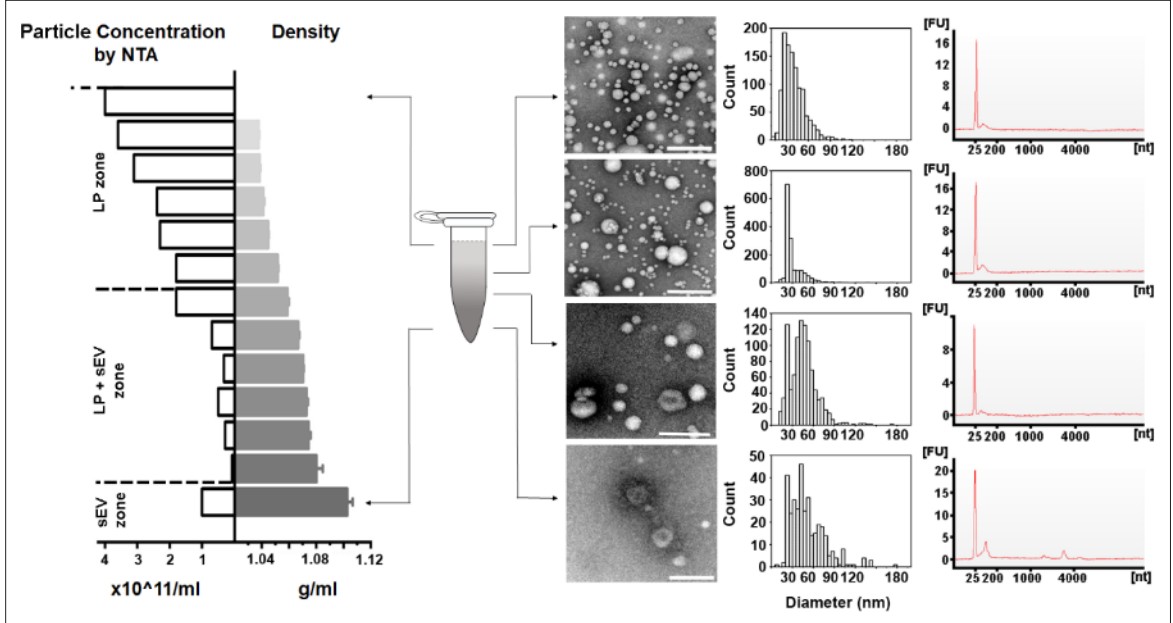

**Figure 5.** Density gradient ultracentrifugation (DGUC) in the 1.5 mL tube format effectively separated small extracellular vesicles (sEVs) from lipoproteins. Particle fractions (PFs) obtained from size exclusion chromatography (SEC) were concentrated into 500 µL, loaded onto the density gradient, and subjected to ultracentrifugation as previously described. After DGUC, the 1.5 mL tube was fractionated into 13 fractions, which were each examined for their particle concentration (by nanoparticle tracking analysis [NTA]) and the presence of sEVs and lipoproteins (by transmission electron microscopy [TEM]). In the sEV zone (bottom of the tube), where density was higher than 1.08 g/mL, high-purity sEVs were indeed observed. A mixed population of sEVs and lipoproteins was observed within the density zone of 1.05–1.08 g/mL. Interestingly, the particle numbers in this density zone were low, thus creating an effective separation zone between lipoproteins and sEVs. The RNA profiles of corresponding particle populations are shown on the right. All scale bars represent 200 nm.

The online version of this article includes the following source data and figure supplement(s) for figure 5:

**Source data 1.** Numerical data corresponding to particle concentration profiles, particle size distributions, and RNA electropherogram profiles.

**Figure supplement 1.** Transmission electron microscopy (TEM) images and size distributions of the 13 fractions collected from the 1.5 mL tube following size exclusion chromatography (SEC)-density gradient ultracentrifugation (DGUC).

**Figure supplement 1—source data 1.** Numerical data corresponding to transmission electron microscopy (TEM)-derived particle size distributions for 13 fractions.

**Figure supplement 2.** SEC-DGUC fractionation reveals efficient sEV isolation after 2h ultracentrifugation.

**Figure supplement 2—source data 1.** Numerical data corresponding to particle size distributions and particle concentration profiles under different centrifugation durations.

Interestingly, when the ultracentrifugation time was increased to 16 hr, the size histogram and particle concentration profile closely mirrored those of the 2 hr DGUC (*Figure 5—figure supplement 2C–E*). This observation suggests that a 2 hr DGUC was sufficient to separate sEVs from lipoproteins in the 1.5 mL tube using the fixed-angle rotor.

In conclusion, the SEC-DGUC protocol successfully isolated sEVs from a small volume of plasma under 3 hr in total by leveraging the strengths of SEC and DGUC.

## DGUC-SEC protocol also isolated high-purity sEVs

An alternative approach to the sequential SEC-DGUC process involves reversing the order, implementing DGUC prior to SEC (termed DGUC-SEC) (*Karimi et al., 2018*; *Onódi et al., 2018*). DGUC separates high-density particles from low-density ones in the plasma, while SEC further differentiates proteins and HDL from sEVs. Theoretically, the DGUC-SEC protocol should also effectively isolate sEVs from plasma (*Holcar et al., 2020*). However, it remains intriguing to assess if the sequence of applying SEC and DGUC impacts the quality of sEV isolates from a small volume of plasma.

For this purpose, 500 µL of plasma was initially processed through a 2 hr DGUC protocol using the 1.5 mL tube format. The bottom 120 µL high-density fraction (plasma-DGUC-1) was then collected

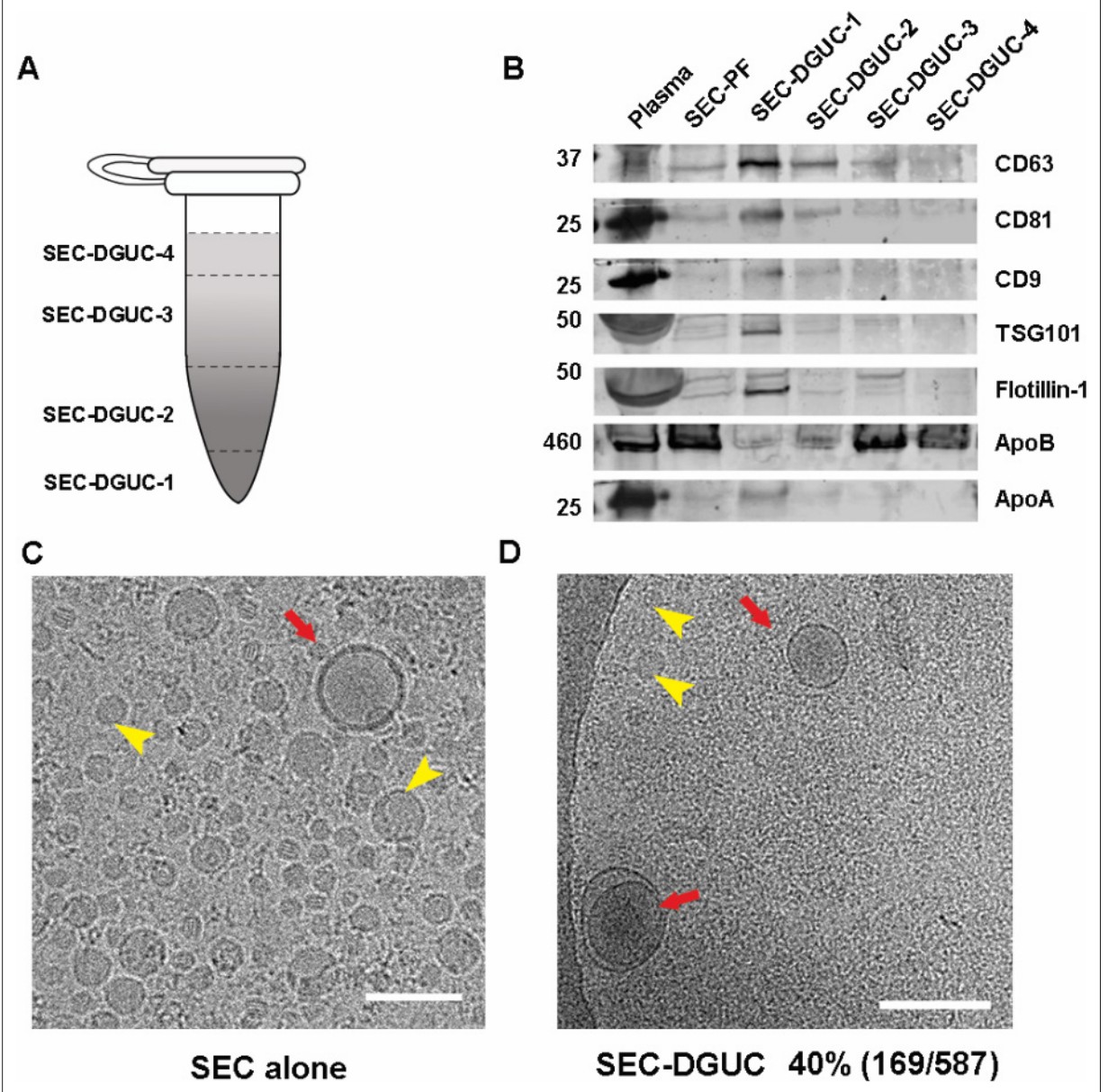

**Figure 6.** The high purity of small extracellular vesicles (sEVs) in SEC-DGUC-1 was demonstrated by WB and cryo-EM. (**A**) The 1.5 mL tube subjected to the size exclusion chromatography (SEC)-density gradient ultracentrifugation (DGUC) protocol was fractionated into four fractions for easier analysis. (**B**) Western blot analyses of original plasma, SEC-PF, SEC-DGUC-1–4 using sEV and lipoprotein markers. (**C**, **D**) Cryo-EM images showing the comparison between SEC-PF and SEC-DGUC-1. The SEC-DGUC-1 in (**D**) was obtained from non-fasting plasma collected in ethylenediaminetetraacetic acid (EDTA) tubes. All scale bars represent 200 nm.

The online version of this article includes the following source data and figure supplement(s) for figure 6:

**Source data 1.** Original uncropped western blot images for *Figure 6*.

**Source data 2.** Annotated uncropped western blot images for *Figure 6*, indicating lane identities and bands used in the analysis.

**Figure supplement 1.** Additional cryo-EM images of SEC-DGUC-1.

and subjected to SEC (Materials and methods). Similar to the SEC-DGUC protocol, this DGUC-SEC protocol also yielded a high-purity sEV isolate, termed DGUC-SEC-PF, as confirmed by western blot and TEM (*Figure 7A and D*).

## Comparison of SEC-DGUC and DGUC-SEC protocols: SEC-DGUC obtained higher yield of sEVs

A side-by-side comparison of sEV isolates from the same plasma source using SEC-DGUC and DGUC-SEC was carried out using several independent characterization methods: western blot, TEM, and total RNA analysis (*Figure 7*, see Materials and methods). In the western blot, an additional sEV sample isolated by a routine dUC protocol was included for comparison. Both sEV samples isolated by the combination of SEC and DGUC using the same amount of plasma showed low contaminants (Calnexin, Albumin, ApoA, and ApoB) and clear signal of sEV markers such as CD9, CD63, CD81, and TSG101 (*Figure 7A*). In stark contrast, sEV isolated by dUC showed a high degree of contamination of Albumin, ApoA, and ApoB, but also an abundance of CD9, CD81, and TSG101. TEM images demonstrated similar appearances of the particles isolated by the two protocols. The particle size distributions of isolated sEVs obtained from the TEM images by both SEC-DGUC and DGUC-SEC protocols also largely overlapped with each other (*Figure 7B*), further suggesting that the particles isolated by both protocols are similar both in chemical and in physical aspect. Despite the similarity of SEC-DGUC-1 and DGUC-SEC-PF samples, the SEC-DGUC-1 showed higher signal intensity for all four tested sEV markers (CD9, CD63, CD81, and TSG101), with estimated concentrations approximately 2.1, 2.1, 4.7, and 4.2 times higher compared to the DGUC-SEC-PF in the western blot. In addition, the total RNA analysis (*Figure 7E and F*) showed that SEC-DGUC-1 contained more than 4 times the total amount of RNA than DGUC-SEC-PF, suggesting that the SEC-DGUC protocol yielded significantly more sEVs.

In conclusion, both the SEC-DGUC and DGUC-SEC protocols proved effective in isolating high-purity sEVs from plasma. Yet, in the context of our experiment, which aimed to isolate sEVs from low plasma volumes, the SEC-DGUC protocol outperformed DGUC-SEC, yielding a significantly greater quantity of sEVs.

## High purity of sEVs facilitates downstream analyses

Building upon the results of the comparative analyses, we further examined the sEVs isolated by SEC-DGUC through additional downstream analyses—surface marker detection by FCM and protein content analysis by liquid chromatography with tandem mass spectrometry (LC-MS/MS).

The sEV surface proteins were analyzed by FCM using a commercial kit such as the MACSPlex Exosome Kit, in a semiquantitative manner (*Figure 8A and B*, see Materials and methods). A panel of 37 surface markers comprehensively evaluated the sEVs in the plasma concerning their origin and relative amount.

We compared the surface marker signals of SEC-DGUC-1 with SEC-PF (*Figure 8C*), derived from the same plasma source, using the same amount of particles ($5 \times 10^8$, calculated from the NTA data), which was reported to be a saturating number (*Wiklander et al., 2018*). The surface marker panel demonstrated striking differences in signal strength between SEC-DGUC-1 and SEC-PF. For some highly expressed surface markers, such as CD42a, CD41b, and CD29, the signals of SEC-DGUC-1 were up to three orders of magnitude higher than SEC-PF (*Figure 8C*). Moreover, for other surface markers such as CD3, CD19, HLAs, CD44, CD45, CD31, and SSEA-4, SEC-DGUC-1 provided much higher signals than the background, compared to SEC-PF, which showed undetectable levels of signals. sEV isolates from two more different plasma exhibited a similar trend (*Figure 8—figure supplement 1*). The contrast between signals from SEC-DGUC-1 and SEC-PF further confirmed the high purity of sEVs in SEC-DGUC-1 and suggested that high-purity sEV isolates would lead to higher sensitivity in the FCM analysis.

Another crucial question to address is whether the sEVs isolated by SEC-DGUC represent the sEV population in the original plasma. To investigate this, we compared the signal patterns of the 37 surface markers across plasma, SEC-PF, SEC-DGUC-1, and sEVs isolated by dUC (*Figure 8D*). The signal pattern was calculated by normalizing individual surface marker signals to their corresponding signal sum. The number of particles loaded for plasma, SEC-PF, SEC-DGUC-1, and sEV isolated by dUC were $1 \times 10^{10}$, $1 \times 10^{10}$, $5 \times 10^8$, and $5 \times 10^8$ (based on NTA measurement), respectively. The number

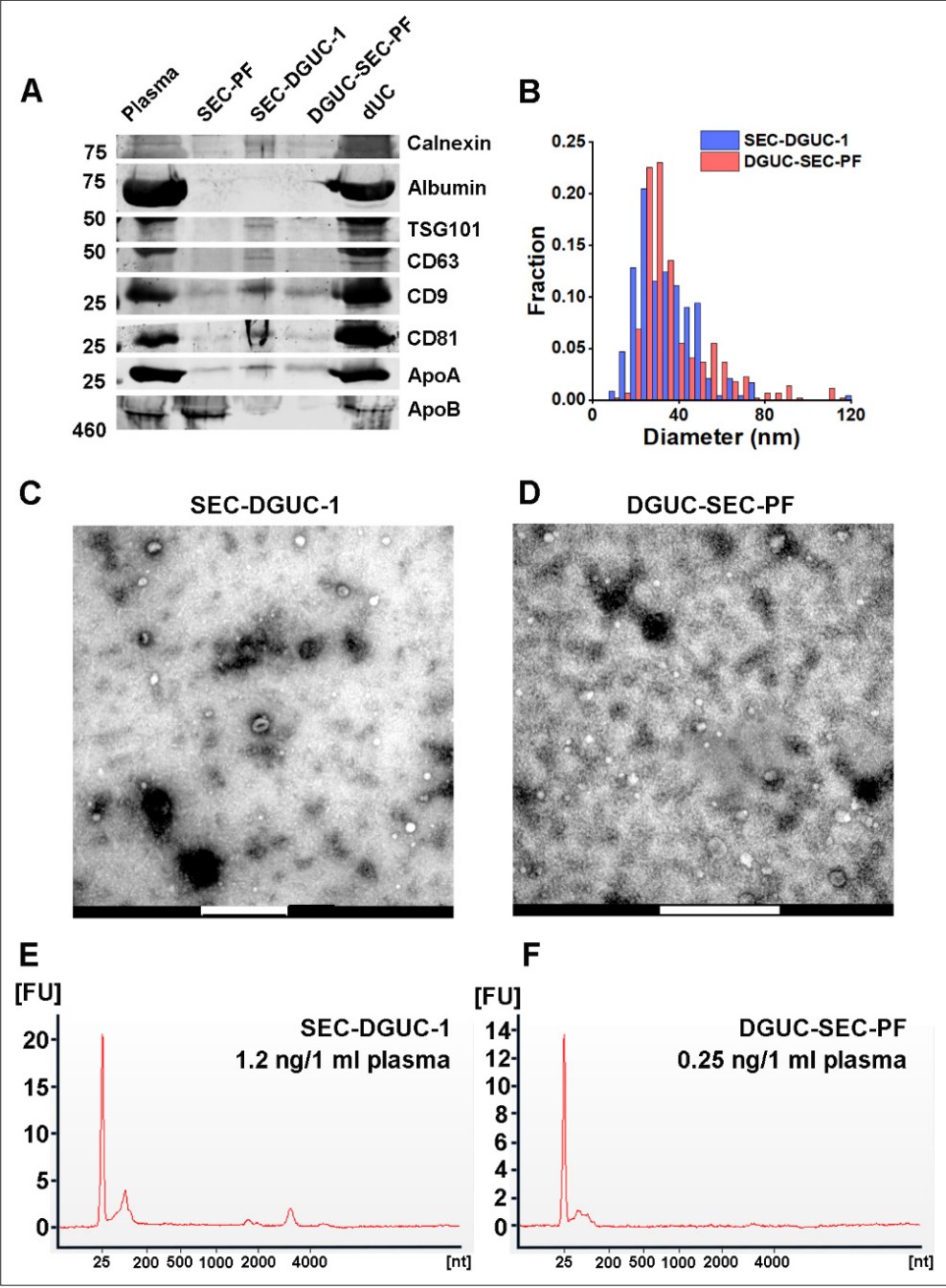

**Figure 7.** Comparison of small extracellular vesicles (sEVs) isolated by size exclusion chromatography (SEC)-density gradient ultracentrifugation (DGUC) and DGUC-SEC protocols. (**A**) Western blot of SEC-DGUC-1, DGUC-SEC-PF, sEV isolated by differential ultracentrifugation (dUC) obtained from 2 mL of plasma of the same source. (**B**) Particle size distributions in SEC-DGUC-1 and DGUC-SEC-PF measured by transmission electron microscopy (TEM). (**C**, **D**) TEM images of SEC-DGUC-1 and DGUC-SEC-PF. (**E**, **F**) Total RNA analyses of SEC-DGUC-1 and DGUC-SEC-PF. Note that data in (**E**) is the same data shown in the bottom right subfigure of *Figure 5*.

The online version of this article includes the following source data for figure 7:

**Source data 1.** Numerical data corresponding to particle diameter distributions derived from transmission electron microscopy (TEM) images and RNA electropherogram profiles.

**Source data 2.** Original uncropped western blot images for *Figure 7*.

**Source data 3.** Annotated uncropped western blot images for *Figure 7*, indicating lane identities and bands used in the analysis.

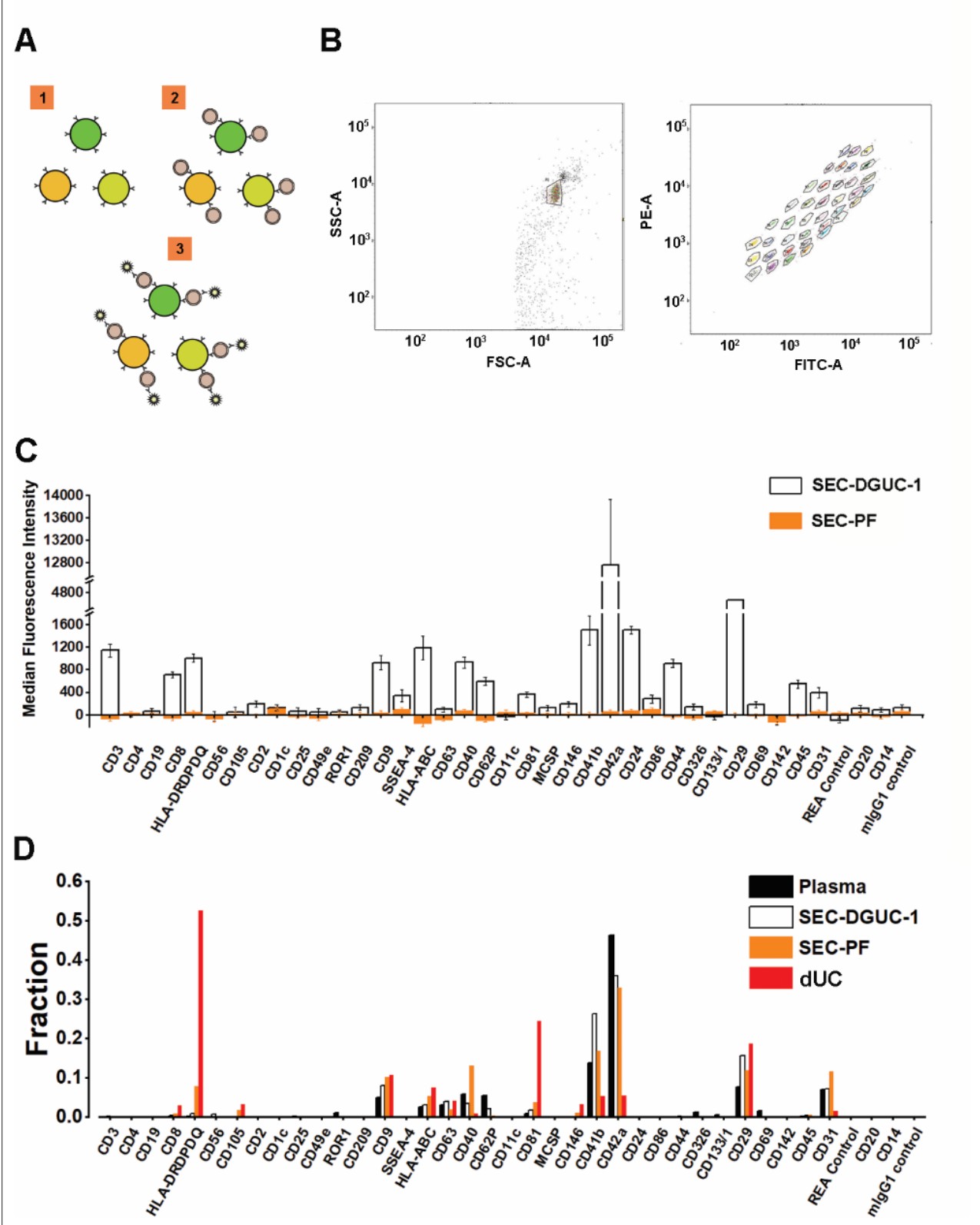

**Figure 8.** Flow cytometry assay using MACSPlex Exosome Kit to evaluate small extracellular vesicle (sEV) isolates. (**A**) Illustration of the principle of MACSPlex Exosome Kit. 37 types of beads with different fluorescent colors are functionalized with specific antibodies against sEV surface proteins. sEVs captured by the beads are detected with flow cytometry using APC-labeled anti-CD9, CD63, CD81 antibodies. (**B**) Flow cytometry gating setup. (**C**) A representative data showing the comparison of median fluorescence intensity (MFI) ± standard error of the mean (SEM) from sEV isolates obtained

*Figure 8 continued on next page*

*Figure 8 continued*

from SEC-PF and SEC-DGUC-1. Equal numbers of particles ($5\times10^8$, based on nanoparticle tracking analysis [NTA]) were loaded for both samples. (**D**) Comparison of signal patterns among plasma, SEC-DGUC-1, SEC-PF, and differential ultracentrifugation (dUC). The results showed that sEVs isolated from dUC deviate from plasma, whereas sEVs in the SEC-PF and SEC-DGUC-1 mirror the sEV population in plasma.

The online version of this article includes the following source data and figure supplement(s) for figure 8:

**Source data 1.** Numerical data corresponding to flow cytometry analysis of surface marker expression.

**Figure supplement 1.** Flow cytometry data of size exclusion chromatography (SEC)-particle fraction (PF) vs. SEC-DGUC-1 obtained from two plasma sources using MACSPlex Exosome Kit.

**Figure supplement 1—source data 1.** Numerical data corresponding to flow cytometry analysis across plasma samples.

of particles loaded for plasma and SEC-PF was 20 times more than SEC-DGUC-1 to achieve a comparable level of sEV loading. The surface marker signal patterns of SEC-PF and SEC-DGUC-1 were similar to plasma, whereas sEVs from dUC deviated from the plasma (***Figure 8D***). This suggested that the sEVs isolated by the SEC-DGUC protocol represented the sEV population in plasma, whereas sEVs isolated by dUC represented a biased population.

In summary, SEC-DGUC-1 significantly improved the sensitivity of surface marker detection compared to SEC-PF in FCM, and the sEVs in SEC-DGUC-1 were found to be representative of the sEV population in the original plasma.

Next, the SEC-PF and SEC-DGUC-1 obtained from 5 mL of a single plasma source were analyzed by LC-MS/MS (Materials and methods). 342 proteins were identified in SEC-PF, and 753 proteins were identified in SEC-DGUC-1. Out of the 754 proteins identified in SEC-DGUC-1, 51 proteins matched with the top 100 sEV proteins from Vesiclepedia. Specific sEV-associated proteins such as CD81, CD9, and TSG101 were identified. Additionally, other common EV-associated proteins such as 14-3-3 protein (theta, zeta/delta), Rabs (Rab-7A and Rab-8A), ADAM10, EHD4, PFN1, immune system-related proteins (MHC Class II), MSN, SDCBP, ENO1, Annexin, GAPDH, ACTN4, TUBA1C, and ESCRT accessory (Clathrin) were also identified. The following cell-specific markers were detected: CD41, CD36, CD42b for platelets; CD37 for leukocytes, and CD163 for macrophages. Regarding the co-isolated contaminating lipoproteins in the sEV isolate, only ApoA was detected in the top 50 most abundant proteins according to emPAI (***Supplementary file 1a***), which was expected based on the western blot result (***Figure 6B***). On the other hand, for SEC-PF, 7 out of the top 11 most abundant proteins were lipoproteins (***Supplementary file 1b***). The identified proteins were further analyzed with the functional annotation tool on the platform of the Database for Annotation, Visualization, and Integrated Discovery (DAVID v6.842), of which the top 10 in the functional annotation chart were shown

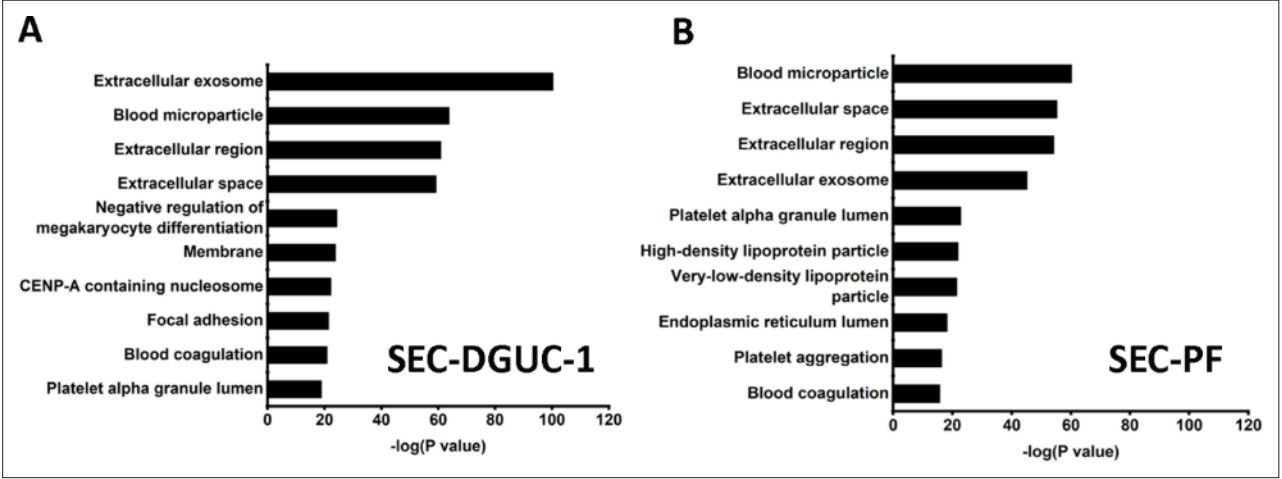

**Figure 9.** Functional annotation of proteins identified by liquid chromatography with tandem mass spectrometry (LC-MS/MS). (**A**) Functional annotation of proteins identified in size exclusion chromatography (SEC)-DGUC-1. (**B**) Functional annotation of proteins identified in SEC-particle fraction (PF).

The online version of this article includes the following source data for figure 9:

**Source data 1.** Numerical data corresponding to functional annotation and enrichment analysis of proteins identified by liquid chromatography with tandem mass spectrometry (LC-MS/MS).

in *Figure 9*. Extracellular exosome was the top first group for SEC-DGUC-1 (*Figure 9A*), suggesting the identified proteins were highly associated with sEVs. In comparison, the top 3 annotation functions of proteins identified in SEC-PF were not related to sEVs (*Figure 9B*).

In summary, sEVs isolated from a small volume of plasma using the SEC-DGUC protocol can provide the required purity to identify sEV-associated proteins in mass spectrometry.

## Repeatability and reliability of the SEC-DGUC protocol

We subsequently assessed the repeatability and reliability of the SEC-DGUC protocol in sEV isolation from plasma, critically analyzing variations at each step of the protocol.

The repeatability of the SEC step was examined using multiple SEC-PFs derived from the same plasma source. Comparisons were based on particle concentration and size distribution (via NTA) once concentrated to 500 μL. This test, utilizing six distinct plasma sources, revealed that the coefficient of variation (CV) in particle concentration ranged between 6% and 55% (*Figure 10—figure supplement 1* table). Meanwhile, size distributions exhibited no notable variations across plasma samples (*Figure 10—figure supplement 1A–C*), implying that despite potential variations in particle numbers, the particle populations remained largely consistent across PFs.

The complete SEC-DGUC protocol was then scrutinized for its repeatability using multiple sEV isolates (SEC-DGUC-1) from the same plasma source. Particle concentrations and size distributions were evaluated, yielding CVs of 24% and 25% for particle concentration in SEC-DGUC-1 (*Figure 10—figure supplement 1* table). We then compared size distributions for each plasma fraction using Jensen-Shannon divergence (JSD). The JSD values, which are well below 0.1 (*Figure 10B*), indicate a consistent population of isolated particles, as further supported by *Figure 10—figure supplement 2*. The repeatability of the SEC-DGUC protocol was further evidenced by its consistent particle concentration profile across the 1.5 mL tube. Four technical replicates of the profiles showed good repeatability (*Figure 10A*), suggesting the SEC-DGUC protocol's ability to robustly generate a consistent density gradient profile and particle subpopulations. Additionally, NTA-measured size distributions displayed well-overlapped histograms of particles (*Figure 10B*), reinforcing the protocol's robustness.

In the subsequent phase, we gauged the protocol's reliability across various plasma samples by evaluating particle concentration profiles, size distributions, and TEM images of SEC-DGUC-1. Particle concentration profiles derived from diverse plasma samples displayed a signature 'tick' shape, indicative of reliable sEV and lipoprotein separation irrespective of the plasma source (*Figure 11—figure supplement 1*). However, the number of particles (*Table 1*) and size distributions of SEC-DGUC-1 varied across different plasma samples (*Figure 11—figure supplement 2*), suggesting that the characteristics of sEVs were plasma-specific. TEM images of SEC-DGUC-1 derived from five different biobanked plasma samples further confirmed the high purity of sEVs, with many particles exhibiting the characteristic cup shape and a low contrast (*Figure 11*).

In conclusion, the SEC-DGUC protocol demonstrated a CV of ~25% for particle number repeatability when isolating sEVs from the same plasma source. Despite this, it exhibited a reliable capacity to isolate high-purity sEVs from varying plasma sources.

## Discussion

The task of isolating sEVs from human plasma becomes particularly challenging when dealing with minimal volumes. To tackle this challenge, our study scrutinized two commonly employed methods, SEC and DGUC, and further investigated the efficiency of their combined use. We found that neither SEC nor DGUC, when used individually, provide an optimal solution for sEV isolation. SEC, although effective in removing plasma proteins, results in a PF dominated by lipoproteins (99%). DGUC, on the other hand, facilitates substantial particle and protein removal due to its density-based separation; however, it does not completely eliminate protein contamination.

The joint use of these methods has been explored previously, yet for small plasma volumes, understanding how to strategically integrate these techniques to maximize efficiency and sEV purity is crucial. Our devised SEC-DGUC protocol merges the advantages of both techniques, employing SEC to eliminate plasma proteins and HDL, and DGUC to remove remaining lipoproteins.

Interestingly, our results showed that performing SEC prior to DGUC yields a higher yield of sEVs, especially for plasma volumes ranging from 500 μL to 2 mL. The gel matrix of the SEC column tends

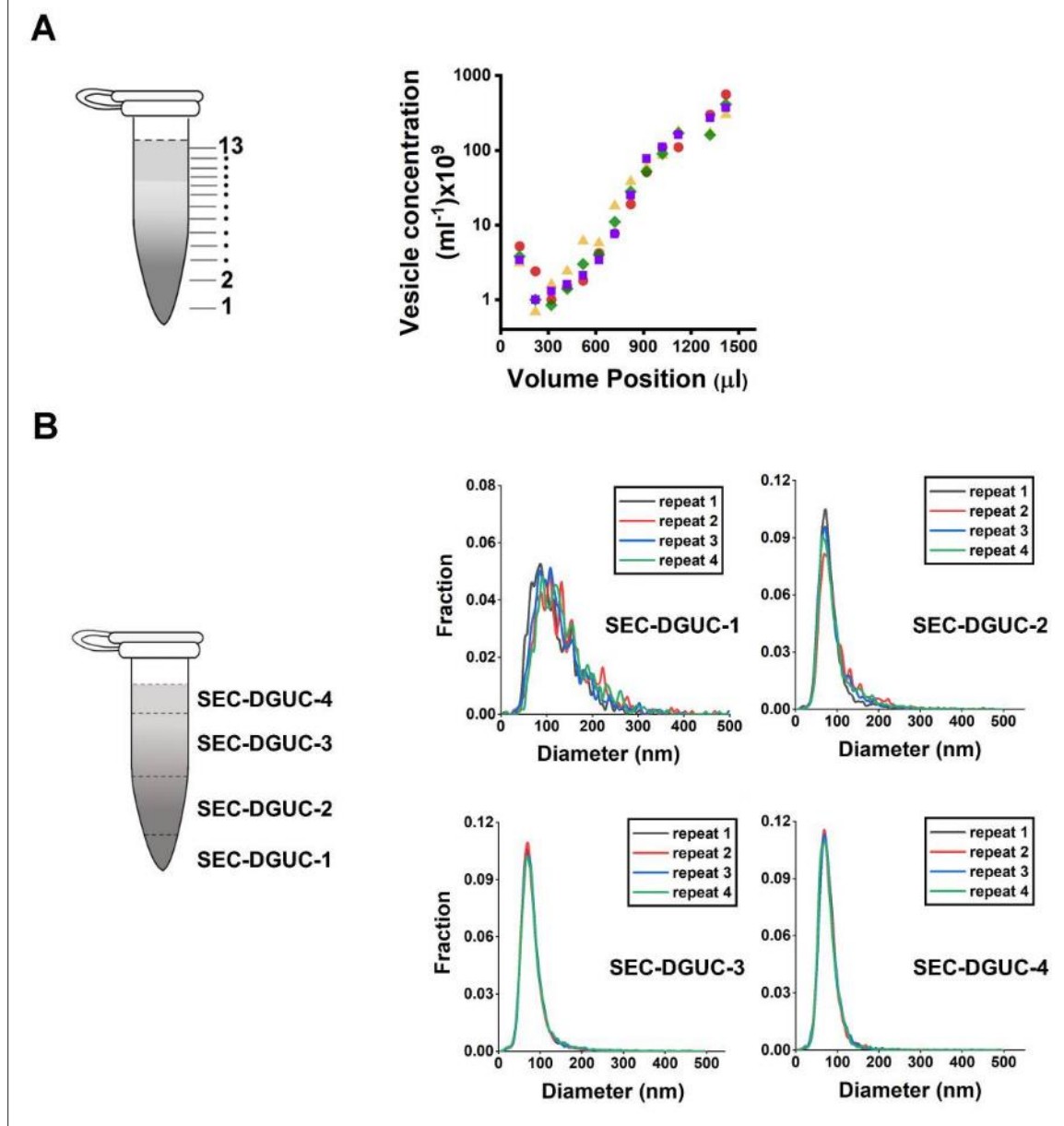

**Figure 10.** Repeatability of the size exclusion chromatography (SEC)-density gradient ultracentrifugation (DGUC) protocol. (**A**) Particle concentration profiles (by nanoparticle tracking analysis [NTA]) along the 1.5 mL tube from four technical replicates subjected to SEC-DGUC protocol. (**B**) Particle size distributions measured by NTA in the four fractions indicated on the left figure. Four technical replicates subjected to SEC-DGUC protocol were shown. Jensen-Shannon divergence (JSD) values for SEC-DGUC-1–4 are 0.015, 0.006, 0.001, and 0.002, indicating strong similarities among the histograms.

The online version of this article includes the following source data and figure supplement(s) for figure 10:

**Source data 1.** Numerical data corresponding to particle concentration profiles and size distributions from repeatability experiments.

**Figure supplement 1.** Repeatability of size exclusion chromatography (SEC).

**Figure supplement 1—source data 1.** Numerical data corresponding to particle size distributions and summary statistics from repeatability experiments.

**Figure supplement 2.** Repeatability of size exclusion chromatography (SEC)-density gradient ultracentrifugation (DGUC) protocol.

**Figure supplement 2—source data 1.** Numerical data corresponding to particle size distributions from repeatability experiments.

**Table 1.** Particle numbers in different steps during the size exclusion chromatography (SEC)-density gradient ultracentrifugation (DGUC) protocol, calculated based on nanoparticle tracking analysis (NTA) measurement.

The first column provides the total particle numbers measured in 500 µL fasting plasma (~1 mL of whole blood) from five individuals. The second column provides the corresponding total vesicle numbers in SEC-particle fraction (PF) after the plasma was subjected to SEC. The third column provides the corresponding total vesicle numbers in SEC-DGUC-1 after the SEC-PF was subjected to DGUC.

| Plasma (particle number) | SEC-PF (particle number) | SEC-DGUC-1 (particle number) |
|---|---|---|
| $8\times10^{10}$ | $6.5\times10^{10}$ | $1.8\times10^{9}$ |
| $5.5\times10^{11}$ | $1.65\times10^{11}$ | $1.32\times10^{9}$ |
| $1.75\times10^{11}$ | $8.5\times10^{10}$ | $1.68\times10^{9}$ |
| $6\times10^{11}$ | $1.2\times10^{11}$ | $1.56\times10^{9}$ |
| $2.4\times10^{10}$ | $1.6\times10^{10}$ | $1.2\times10^{9}$ |

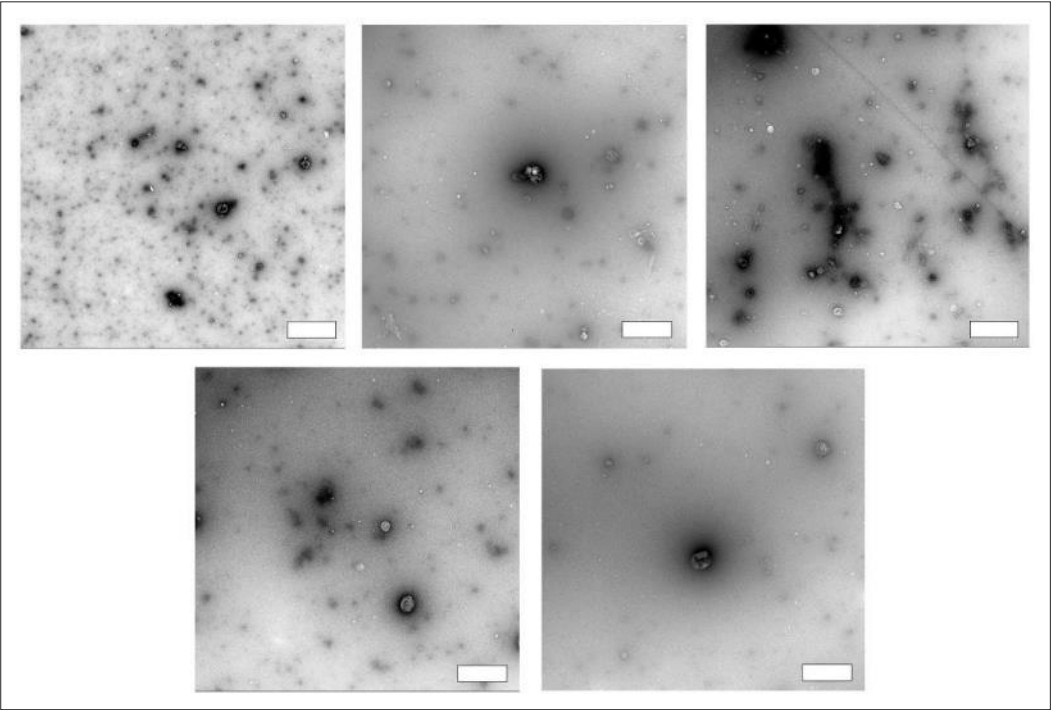

**Figure 11.** Reliability of the size exclusion chromatography (SEC)-density gradient ultracentrifugation (DGUC) protocol. Transmission electron microscopy (TEM) images of SEC-DGUC-1 were obtained from five fasting plasma (biobank samples, ethylenediaminetetraacetic acid [EDTA] tubes). High purity of small extracellular vesicles (sEVs) represented by the low-contrast, cup-shaped vesicles was evident across different samples, suggesting the SEC-DGUC was reliable in obtaining high-purity sEVs. All scale bars represent 200 nm.

The online version of this article includes the following source data and figure supplement(s) for figure 11:

**Figure supplement 1.** Reliability of particle concentration profiles along the 1.5 mL tube (by nanoparticle tracking analysis [NTA]).

**Figure supplement 1—source data 1.** Numerical data corresponding to particle concentration profiles across fractions for multiple plasma samples.

**Figure supplement 2.** Size distributions (by nanoparticle tracking analysis [NTA]) of size exclusion chromatography (SEC)-DGUC-1 obtained from five fasting plasma corresponding to *Figure 11*.

**Figure supplement 2—source data 1.** Numerical data corresponding to particle size distributions of size exclusion chromatography (SEC)-DGUC-1 across plasma samples.

to retain a portion of proteins and particles, causing sample loss. When a low particle concentration sample is processed through an SEC column, this loss becomes more significant. Consequently, performing SEC first—when sEVs, lipoproteins, and plasma proteins are all present—distributes the loss among all components. Given the minor proportion of sEVs in plasma, the loss is more likely absorbed by lipoproteins and plasma proteins. Conversely, protocols that use SEC in the final isolation step could experience a significant loss of sEVs, as most lipoproteins and plasma proteins have already been removed (*Zhang et al., 2020*). Such protocols may yield a lower amount of sEVs or require a larger plasma volume to obtain enough sEVs for subsequent analyses (*Zhang et al., 2020*; *Karimi et al., 2018*). This could explain the lower yield of sEVs observed in the DGUC-SEC protocol compared to the SEC-DGUC protocol.

The SEC-DGUC strategy has been adopted by others and has shown reliable results (*Vergauwen et al., 2021*). However, DGUC is typically performed in tubes of larger volume capacity, such as 12 mL (*Vergauwen et al., 2021*) or 16.8 mL (*Karimi et al., 2018*) placed in swinging-bucket rotors. For our SEC-DGUC protocol, the DGUC was carried out in a 1.5 mL tube format, and the density gradient setup was designed to harvest all sEVs with a density greater than 1.08 g/mL, rather than in a specified narrow range of density. From our experiments, we recognized that small tubes are easier to handle, require shorter handling time, as well as lower volumes of reagents, are more suitable for benchtop ultracentrifuges, and the fractions can be harvested manually without the need of specialized equipment. More importantly, DGUC with the small-tube format is done using a fixed-angle rotor, which has a shorter sedimentation path length. This requires a much shorter time to isolate sEVs (<3 hr) compared to a larger volume tube format using a swinging-bucket rotor, which typically lasts ~16 hr or more (*Vergauwen et al., 2021*). Moreover, DGUC using the small-tube format resulted in high-purity sEV isolates collected in a volume of 120 µL, the concentration of which enabled direct downstream analyses such as electron microscopy, western blot, total RNA analysis, and a multiplex bead-based flow cytometric assay.

The advantage of high-purity sEV isolates was demonstrated by FCM and mass spectrometry. FCM results demonstrated that the sEVs isolated by SEC-DGUC gave a signal three orders of magnitude higher than using SEC alone, thereby improving the sensitivity of the assay to a greater extent. Moreover, the surface marker pattern of 37 proteins helps to assess the nature of the sEV isolates. The surface marker patterns matched closely among the sEVs isolated from SEC-DGUC, SEC alone, and the original plasma, suggesting a close representation of sEVs isolated from SEC-DGUC to the original plasma. It is important to note that the surface marker patterns of sEVs isolated from a routine dUC protocol deviated significantly from sEV populations in the original plasma.

There have been several studies using mass spectrometry to determine the proteome of plasma-derived EVs (*Karimi et al., 2018*; *de Menezes-Neto et al., 2015*; *Looze et al., 2009*; *Karimi et al., 2022*). *Karimi et al., 2018*, analyzed sEVs isolated by a DGUC-SEC protocol from large volumes of pooled healthy human plasma with LC-MS/MS and identified 1187 proteins. Many proteins identified were sEV-associated proteins such as CD63, CD9, and CD81, and the identified proteins were closely associated with extracellular exosomes. In comparison, when analyzing the sEVs isolated by the newly introduced SEC-DGUC protocol using 5 mL plasma from a single healthy individual, the proteomics data identified 753 proteins, which included sEV-specific proteins, such as CD81, CD9, Flotillin, and TSG101. Moreover, the proteins were closely associated with extracellular exosomes as well. We expect that, with a better optimized LC-MS/MS protocol, the volume of plasma required for the sEV isolation can be further reduced. Therefore, we conclude that the sEVs isolates obtained using the small-volume plasma SEC and DGUC protocol are of sufficient yield and purity for mass spectrometry analysis.

The key to the consistency of the SEC-DGUC protocol lies in three aspects: first, the capability of SEC to efficiently remove plasma proteins and HDL; second, the capacity of DGUC to separate sEVs from low-density lipoproteins; third, the careful manual handling during DGUC, especially in the final step of harvesting the sEVs from the tube. When evaluated individually, SEC is a well-established and quick method which is commercially available; DGUC with the simple density gradient design enables consistency, precision, and the potential to fully automate the protocol, thereby mitigating human error. For these reasons, the SEC-DGUC protocol can be extrapolated to obtain high-purity sEVs in a repeatable and reliable manner.

Collectively, our results clearly demonstrated the quality of the isolated sEVs and the robustness of the SEC-DGUC protocol in a small-tube format. This is not to claim that the existing protocols are less reliable than the currently developed protocol. In fact, there is room to further improve the protocol in future efforts.

The SEC-DGUC protocol, though efficient, has limitations like potential damage to or aggregation of sEVs (*de Menezes-Neto et al., 2015*) and low-level contamination with HDL. The iodixanol reagent introduced in DGUC could potentially interfere with certain downstream analyses, and variations in plasma samples might impact sEV purity. Although the protocol effectively isolates representative sEVs from plasma, some lower density sEVs that require a longer centrifugation time may be missed.

Isolating high-purity sEVs from small volumes of plasma is essential in fully realizing the potential of using sEVs as biomarkers. It is of particular importance to standardize the sEV isolation protocol so that the data obtained by different labs can be compared and integrated with high confidence. The SEC-DGUC protocol presented here is relatively short, easy to carry out, and isolates sEVs with high purity in a repeatable and reliable manner. The SEC-DGUC protocol can be quickly adapted by any laboratory and can aid clinical studies that have limited accessible plasma volumes. It potentially offers a pathway for the standardization of sEV isolation from plasma.

# Materials and methods

## Key resources table

| Reagent type (species) or resource | Designation | Source or reference | Identifiers | Additional information |
|---|---|---|---|---|
| Commercial assay or kit | ACD-A vacuette tube | Greiner, Austria | 455055 | |
| Commercial assay or kit | SEC columns | HansaBioMed Life Sciences, Tallinn, Estonia | HBM-PEV | 5 mL capacity |
| Chemical compound, drug | Iodixanol | Sigma-Aldrich | OptiPrep, D1556 | |
| Other | Ultracentrifuge tube | Beckman Coulter | 357448 | |
| Commercial assay or kit | BCA protein assay kit | Pierce, Thermo Fisher | 23225 | |
| Other | ZetaView | Particle Metrix | PMX 120 | |
| Antibody | Mouse monoclonal anti-CD63 | Santa Cruz Biotechnology, Dallas, TX, USA | sc-5275; clone MX-49.129.5 RRID:AB_627877 | WB (1:1000) |
| Antibody | Rabbit monoclonal anti-CD9 | Abcam, Cambridge, MA, USA | ab92726; clone 9EPR2949 RRID:AB_10561589 | WB (1:1000) |
| Antibody | Mouse monoclonal anti-CD81 | Santa Cruz Biotechnology, Dallas, TX, USA | sc-166029; clone B-11 RRID:AB_2275892 | WB (1:1000) |
| Antibody | Rabbit polyclonal anti-Flotillin-1 | Abcam, Cambridge, MA, USA | ab41927 RRID:AB_941621 | WB (1:1000) |
| Antibody | Rabbit polyclonal anti-TSG101 | Abcam, Cambridge, MA, USA | ab30871 RRID:AB_2208084 | WB (1:1000) |
| Antibody | Mouse monoclonal anti-ApoA-I | Santa Cruz Biotechnology, Dallas, TX, USA | sc-376818; clone B-10 RRID:AB_2797313 | WB (1:1000) |
| Antibody | Mouse monoclonal anti-ApoB | Santa Cruz Biotechnology, Dallas, TX, USA | sc-13538; clone C1.4 RRID:AB_626690 | WB (1:1000) |
| Antibody | Mouse monoclonal anti-Albumin | Abcam, Cambridge, MA, USA | ab10241 RRID:AB_296978 | WB (1:1000) |
| Antibody | Rabbit polyclonal anti-Calnexin | Abcam, Cambridge, MA, USA | ab22595 RRID:AB_2069006 | WB (1:1000) |
| Antibody | IRDye 800CW anti-mouse | Li-COR Biosciences, Lincoln, NE, USA | 925-32210 RRID:AB_2687825 | Secondary antibody, WB (1:15,000) |

*Continued on next page*

*Continued*

| Reagent type (species) or resource | Designation | Source or reference | Identifiers | Additional information |
|---|---|---|---|---|
| Antibody | IRDye 680LT anti-rabbit | Li-COR Biosciences, Lincoln, NE, USA | 926-68021 RRID:AB_10706309 | Secondary antibody, WB (1:15,000) |
| Other | Carbon-coated grid | Electron Microscopy Sciences, Hatfield, PA, USA | CF300-CU | |
| Commercial assay or kit | miRNeasy Serum/Plasma Kit | QIAGEN, GmbH, Hilden, Germany | 217184 | |
| Commercial assay or kit | MACSPlex Exosome Kit | Miltenyi Biotec, Bergish Gladbach, Germany | N/A | |
| Software, algorithm | ImageStudio v5.2 | | N/A | |
| Software, algorithm | Proteome Discoverer v2.1 | Thermo Fisher Scientific | N/A | |
| Other | Xbridge C18 column | Waters, Milford, MA, USA | 4.6 × 250 mm | |

## Blood collection and preparation of platelet-free plasma

All procedures involving peripheral blood specimens were approved by the Singapore National Health Group Domain Specific Review Board (the central ethics committee) and were mutually recognized by the Nanyang Technological University Institutional Review Board (IRB#2018/00671). All blood specimens were de-identified prior to their use in the experiments. The plasma used in this study was obtained from two sources, fresh blood and biobanked plasma. Fresh blood was obtained with written informed consent from healthy adult males aged between 25 and 45 years. All blood was collected after overnight fasting unless specified otherwise in the figure legend. The volunteers were free of medications for at least 3 weeks. Blood was drawn in the morning using venepuncture technique with a 21G butterfly needle into anticoagulant citrate dextrose-A (ACD-A) containing vacuette tubes (455055, Greiner, Austria). Three vacuette tubes (each 9 mL volume) of blood were collected from each volunteer. The vacuette tubes were inverted a few times to mix the blood with the anticoagulant, placed in an upright position, and processed within 20 min of collection. We applied the International Society on Thrombosis and Haemostasis (ISTH) protocol to prepare the platelet-free plasma (PFP). Briefly, the blood was centrifuged at 2500×g for 15 min at room temperature (RT) using a table-top centrifuge (Z206A, HermLe Labortechnik GmbH, Germany) to remove blood cells, resulting in platelet-poor plasma (PPP). The PPP was transferred into a new tube and centrifuged once again at 2500×g for 15 min, resulting in PFP. The PFP was transferred into a new tube, homogenized by gentle inversion, split into 500 µL aliquots in 1.5 mL Eppendorf tubes, and stored at –80°C until further use. The biobanked plasma samples were obtained from healthy males following overnight fasting with written informed consent. Briefly, the blood was collected using venepuncture technique with a 21G butterfly needle into ethylenediaminetetraacetic acid (EDTA) containing vacuette tubes. The blood was then centrifuged at 1200×g for 15 min at 4°C with brake-two in a horizontal swing-bucket centrifuge. The plasma fraction was collected within 0.1 mL of the interphase with the buffy coat layer. The plasma was split into 500 µL aliquots in 1.5 mL Eppendorf tubes and stored at –80°C until further use. All data shown in this paper used ACD-A plasma unless indicated otherwise in the figure legends.

## Size exclusion chromatography

An aliquot of frozen plasma was taken from –80°C and allowed to thaw completely at RT. SEC elution was performed using PURE-EVs SEC columns (HBM-PEV, HansaBioMed Life Sciences, Tallinn, Estonia) following the manufacturer's protocol. Briefly, the SEC columns were equilibrated to RT and washed with 3×10 mL particle-free 1× PBS (SH30256.01, Hyclone, UT, USA) to eliminate preservative buffer residues. A volume of 500 µL of PFP was applied on top of each column. Once the sample was inside the gel matrix, the column was loaded with particle-free 1× PBS (the mobile phase of the SEC column). The column was not allowed to dry out during this process. Each of the 15 fractions of 500 µL volume was collected, after which the column was washed with approximately 20 mL of 1× PBS and stored at 4°C. These SEC columns were washed and reused up to five times.

The first six SEC fractions (3 mL) constituted void volume, hence were discarded, and the rest of the nine fractions were analyzed for particle numbers and protein concentration. Fractions 7–10, which

were subsequently identified as the PF, were pooled to a volume of 2 mL and concentrated using an Amicon Ultra 100 kDa centrifugal filter (Merck Millipore, MA, USA), made from regenerated cellulose, to a final volume of 500 µL.

## Density gradient ultracentrifugation

To prepare the density solutions, a 50% Working Solution was first made by diluting 5 vol of iodixanol (OptiPrep, D1556, Sigma-Aldrich, MO, USA) with 1 vol of 0.25 M sucrose, 6 mM EDTA, 60 mM Tris-HCl (pH 7.4). The Working Solution was then mixed with Homogenization Medium composed of 0.25 M sucrose, 1 mM EDTA, and 10 mM Tris-HCl (pH 7.4) to prepare density solutions of desired percentages.

For DGUC experiment using the 12 mL tube, PFs obtained from 6 mL of plasma (24 mL total volume as previously described) were concentrated to 6 mL and added to a Beckman Coulter ultra-clear centrifuge tube (344059, Beckman Coulter, USA). A density gradient was carefully constructed by sequentially underlaying 2 mL each of 10%, 30%, and 50% iodixanol cushions. The gradient tube was balanced and subjected to ultracentrifugation at 150,000×$g$ for 2 hr at 4°C using a swinging-bucket rotor (SW 41 Ti Beckman Coulter, USA).

For the DGUC using the 1.5 mL tube format, 500 µL of the concentrated PF obtained from a single SEC (as previously described) was added to a Beckman Coulter ultracentrifuge tube (357448, Beckman Coulter, USA). For plasma volume higher than 500 µL, SEC-PFs from multiple SECs were pooled and concentrated to 500 µL before adding to the ultracentrifuge tube. A density gradient was carefully constructed by sequentially underlaying 800 µL of 10% iodixanol solution and 20 µL of 50% iodixanol. The gradient tubes were balanced and subjected to centrifugation at 135,000×$g$ for 2 hr at 4°C using a fixed-angle rotor (TLA-55, Beckman Coulter, USA). After DGUC, 13 fractions starting from the bottom of the tube were manually collected by using gel loading pipetting tips. In total RNA analysis and western blot experiments, fractions 2–4, 5–10, and 11–13 were pooled together for easier analyses. Thus, four main fractions were analyzed, namely: SEC-DGUC-1 (120 µL volume), SEC-DGUC-2 (2–4 pooled, 300 µL volume), SEC-DGUC-3 (5–10 pooled, 600 µL volume), and SEC-DGUC-4 (11–13 pooled, 300 µL volume) (*Figure 6A*).

## DGUC-SEC

For the DGUC-SEC protocol, 500 µL of plasma was added to Beckman Coulter centrifuge tubes (357448, Beckman Coulter, USA), followed by sequentially underlaying 800 µL of 10% iodixanol solution and 20 µL of 50% iodixanol. The gradient tubes were subjected to centrifugation at 135,000×$g$ for 2 hr at 4°C using a fixed-angle rotor (TLA-55, Beckman Coulter, USA). After DGUC, the bottom 120 µL was collected as the high-density fraction and designated as plasma-DGUC-1. The plasma-DGUC-1 was then topped up to a volume of 500 µL with 1× PBS to run through the SEC column. Fractions 7–10 from the SEC were collected, pooled together, and concentrated to a final volume of 120 µL using an Amicon Ultra 100 kDa centrifugal filter (Merck Millipore, MA, USA). For plasma volume higher than 500 µL, multiple tubes were processed in parallel in the DGUC step, and the resulting plasma-DGUC-1 fractions were pooled and subjected to SEC to obtain sEV isolates.

## Total protein analysis

Protein contents of the SEC fractions were measured using a BCA protein assay kit (23225, Pierce, Thermo Fisher Scientific, MA, USA). A standard curve (range 0–1000 µg/mL) was derived with six points of serial dilution with bovine serum albumin (BSA). BSA standard or samples (10 µL) were transferred to a 96-well plate (Thermo Scientific Nunc MicroWell) to which 100 µL working reagent was added (working reagent 50:1 ratio of assay reagents A and B). The plate was incubated for 30 min at 37°C before being analyzed with a multi-well spectrophotometer at 562 nm (Tecan Infinite M200Pro, Switzerland). The average 562 nm absorbance measurement of the blank replicates was subtracted from all other individual standard and unknown sample replicates. A standard curve was obtained from the measurement of BSA standard. The protein concentration of each unknown sample was calculated based on the standard curve.

## Density measurement

The density of fractions obtained from DGUC was determined by absorbance spectroscopy (Tecan Infinite M200Pro, Switzerland) reading at 340 nm in a flat-bottom 96-well polystyrene plate (167008 Nunclon, Thermo Fisher Scientific, MA, USA). A range of iodixanol solutions (5–50%) was diluted at a 1:1 ratio with deionized water and measured in the same 96-well plate to serve as the standard control (*serumwerk.com, 2020*). For iodixanol concentrations above 35%, a second dilution of the solutions was done to avoid absorbance values above 1.2. The absorbance measurements were made against water and 0.25 M sucrose blanks. Each sample was prepared by triplicate to reduce error. Prior to measuring the absorbance, the plate was shaken thoroughly to mix the samples. A standard curve was prepared by plotting the predetermined densities (*Graham, 2002*) of the 2–50% iodixanol solutions against their mean absorbance. Using the standard curve, the densities of the fractions collected from the iodixanol gradient following DGUC were calculated.

## Nanoparticle tracking analysis

NTA was carried out using ZetaView (PMX 120, Particle Metrix, Meerbusch, Germany). Samples were diluted to the recommended particle concentration with particle-free 1× PBS prior to analysis. The particle motion was measured over 11 positions with two repeats at each position with the following parameters: frame rate, 30 fps; number of frames recorded, 60; sensitivity, 80; exposure, 100; minimum brightness, 30; minimum pixel size, 10; maximum pixel size, 10,000; trace length, 30; temperature, 25°C.

## Western blotting

In the first experiment (*Figure 6B*), SEC-PF and sEV isolates from 2 mL of plasma obtained by SEC and SEC-DGUC, respectively, were analyzed for the presence of sEV and lipoprotein markers. The same loading volumes of 22 µL were used for SEC-DGUC-1–4. Since 500 µL SEC-PF was loaded on top of 820 µL iodixanol in the 1.5 mL tube, resulting in a total volume of 1320 µL, the equivalent loading volume for SEC-PF was calculated to be 22×500/1320=8.3 µL. The loading particle number (based on NTA measurement) of SEC-PF and SEC-DGUC-1–4 were $1.8 \times 10^{10}$, $8.8 \times 10^{8}$, $5.9 \times 10^{8}$, $5.5 \times 10^{9}$, $2.86 \times 10^{10}$, respectively. In this manner, not only the amount of sEVs against lipoproteins of each fraction can be evaluated, but also the particle concentration profile in the 1.5 mL tube after SEC-DGUC can be analyzed.

In the second experiment (*Figure 7A*), SEC-PF and three sEV isolates obtained from 2 mL plasma using SEC-DGUC, DGUC-SEC, and dUC were examined for the presence of sEV and lipoprotein markers. The sEVs from SEC-DGUC, DGUC-SEC, and dUC were collected in a volume of 120 µL and the same loading volumes of 30 µL were used. The sample loading volume for SEC-PF (11.4 µL) was calculated as described above.

All samples were lysed in radioimmunoprecipitation assay (RIPA) buffer (89900, Pierce, Thermo Scientific, MA, USA) containing protease inhibitor cocktail (36978, Thermo Scientific, MA, USA) while chilling on ice. After mixing with loading buffer (4×) containing β-mercaptoethanol (80570, Merck, Darmstadt, Germany) and heated to 70°C for 10 min, the samples were loaded onto 12% SDS-PAGE gels and electrophoresed to detect sEV markers (CD63, CD9, CD81, and TSG101, tested in both *Figures 6B and 7A*; Flotillin-1, tested in *Figure 6B* but not in *Figure 7A*), lipoprotein markers (ApoA-I and ApoB), and negative control markers (Calnexin and Albumin, tested in *Figure 7A* but not in *Figure 6B*). After being transferred to 0.45 µm nitrocellulose membranes (1620115, Bio-Rad, Feldkirchen, Germany), proteins were stained with REVERT total protein stain (926-11015, Li-COR Biosciences, Lincoln, NE, USA) for normalization. After this, membranes were blocked at RT for 1 hr with Odyssey blocking buffer TBS (927-50000, Li-COR Biosciences, Lincoln, NE, USA) and incubated overnight at 4°C with 1:1000 of the following antibodies: mouse monoclonal anti-CD63 (MX-49.129.5) (sc-5275, Santa Cruz Biotechnology, Dallas, TX, USA), rabbit monoclonal (9EPR2949) anti-CD9 (ab92726, Abcam, Cambridge, MA, USA), mouse monoclonal anti-CD81 (B-11) (sc-166029, Santa Cruz Biotechnology, Dallas, TX, USA), rabbit polyclonal anti-Flotillin-1 (ab41927, Abcam, Cambridge, MA, USA), rabbit polyclonal anti-TSG101 (ab30871, Abcam, Cambridge, MA, USA), mouse monoclonal anti-ApoA-I (B-10) (sc-376818, Santa Cruz Biotechnology, Dallas, TX, USA), mouse monoclonal anti-ApoB (C1.4) (sc-13538, Santa Cruz Biotechnology, Dallas, TX, USA), mouse monoclonal anti-Albumin (ab10241, Abcam, Cambridge, MA, USA), and rabbit polyclonal anti-Calnexin (ab22595,

Abcam, Cambridge, MA, USA). The membranes were washed by TBS with 0.1% Tween 20 (TBS-T) and incubated at RT for 1 hr with IgG secondary antibodies at 1:15,000, i.e., IRDye 800CW anti-mouse (925-32210, Li-COR Biosciences, Lincoln, NE, USA) and IRDye 680LT anti-rabbit (926-68021, Li-COR Biosciences, Lincoln, NE, USA). Membranes were washed again with 1× TBS-T and scanned with an Odyssey CLx imaging system (Li-COR Biosciences) using 700 and 800 nm channels. Visualization was done by ImageStudio software version 5.2 (Li-COR Biosciences).

## Transmission electron microscopy

The TEM experiments were carried out at the Cryo-Electron Microscopy Platform, NTU Institute of Structural Biology. A carbon-coated grid (CF300-CU, Electron Microscopy Sciences, Hatfield, PA, USA) was glow-discharged for 1 min before placing 4 µL sample adsorption for 1 min. After blotting, the grid was negatively stained for 1 min using 4 µL of 2% uranyl acetate. The grid was then blotted and air-dried for TEM imaging. Grids were imaged using a T12 transmission electron microscope operating at 120 kV with an Eagle 4k HS camera. TEM images were analyzed using ImageJ (*Schneider et al., 2012*) with the Nanodefine plug-in (*Verleysen et al., 2019*) to identify and count particles. Particles with low contrast were difficult to recognize by the Nanodefine plug-in, and they were counted and measured manually using ImageJ measuring tools.

## Cryo electron microscopy

The cryo-EM experiments were carried out at the Cryo-Electron Microscopy Platform, NTU Institute of Structural Biology. SEC-PF and SEC-DGUC-1 samples that demonstrated an NTA concentration of at least $3×10^{11}$ particles were selected for cryo-EM imaging. For SEC-DGUC-1, an additional step of dialysis using the Pur-A-Lyzer Mini 6000 dialysis kit (PURN60030, Sigma-Aldrich, MO, USA) was added to remove iodixanol before grid preparation. The grid preparation protocol follows *Tonggu and Wang, 2020*. Briefly, a volume of 2–3 µL of the sample was applied to a glow-discharged Quantifoil R2/2 grid coated with 2 nm carbon (Jena, Germany) and incubated for 5–10 min. The grid was then loaded into the FEI vitrobot. Another 2 µL of the sample was added to the grid and immediately blotted with the following parameters: blotting time, 3 s; humidity, 100%; temperature, 4°C; blotting force, –1; waiting time, 0 s; and thereafter, grids were flash-frozen in liquid N2-cooled liquid ethane. Grids were imaged on an Arctica transmission electron microscope (FEI) operated at 200 kV on a Falcon III (FEI) direct electron detector. The cryo-EM images were analyzed manually using ImageJ.

## Total RNA isolation

For total RNA isolation, 100 µL isolates obtained from 2 mL of plasma following the SEC-DGUC (SEC-DGUC 1–4) and DGUC-SEC protocols were used. RNA extraction was done using the miRNeasy Serum/Plasma Kit (217184, QIAGEN, GmbH, Hilden, Germany) according to the quick-start protocol by the manufacturer. Briefly, 500 µL of QIAzol lysis reagent (#79306) was added to the isolates, mixed by pipetting, and incubated at RT for 5 min. After adding 100 µL chloroform, the tubes were capped securely and shaken vigorously for 15 s. The tubes were then incubated for 3 min at RT and centrifuged at 12,000×*g* for 15 min at 4°C. The upper aqueous phase (~300 µL), containing total RNA, was carefully transferred to a new 2 mL collection tube without transferring any interphase. 1.5 volumes of 100% ethanol were added (~450 µL), and the samples were mixed thoroughly by pipetting. Up to 700 µL of sample was pipetted into the QIAGEN RNeasy MinElute spin column in a 2 mL collection tube, and the tubes were capped and centrifuged at 8000×*g* for 15 s at RT. The flow-through was discarded. This step was repeated until all the sample passed through the spin column. 700 µL Buffer RWT (QIAGEN #1067933) was added to each column; then the tubes were capped and centrifuged for 15 s at 8000×*g*, RT, after which the flow-through was discarded. 500 µL Buffer RPE (QIAGEN #1018013) was added to each column, the tubes were capped and centrifuged for 15 s at 8000×*g*, RT, then the flow-through was discarded. This step was repeated, but during the second time, the tubes were centrifuged for 2 min instead. The spin column was placed inside a new 2 mL RNAse-free collection tube and was centrifuged at full speed for 5 min at RT with the tube cap left open to dry the membrane. The collection tube containing the flow-through was discarded. Finally, the spin column was placed into a 1.5 mL RNAse-free collection tube, and 14 µL nuclease-free water was added directly to the center of the column, then centrifuged at full speed at RT for 1 min to elute the RNA. The eluate was run through the column again to maximize the RNA recovery. The RNA was

immediately placed on ice. RNA profile and concentration were then assessed by 2100 Bioanalyzer using RNA 6000 Pico kit (Agilent Technologies) according to the manufacturer's protocol.

## sEV surface protein profiling by FCM

The MACSPlex Exosome Kit (Miltenyi Biotec, Bergish Gladbach, Germany) utilizes beads functionalized with 37 types of antibodies to capture sEVs expressing the corresponding surface markers. The sEVs captured by the beads are then labeled with a cocktail of fluorescent-labeled CD9, CD63, and CD81 antibodies, which are detected by FCM. The expression of 37 surface markers (CD1c, CD2, CD3, CD4, CD8, CD9, CD11c, CD14, CD19, CD20, CD24, CD25, CD29, CD31, CD40, CD41b, CD42a, CD44, CD45, CD49e, CD56, CD62P, CD63, CD69, CD81, CD86, CD105, CD133, CD142, CD146, CD209, CD326, HLA-ABC, HLA-DRDPDQ, MCSP, ROR1, and SSEA-4) was evaluated by FCM with the short protocol specified by the manufacturer. Briefly, $5×10^8$ of particles/sample (calculated based on NTA measurement) or PBS (negative control) were diluted with MACSPlex buffer into 1.5 mL microcentrifuge tubes with a final volume of 120 µL. Each tube was incubated at RT for 1 hr in the dark with 10 µL of MACSPlex exosome capture beads and 15 µL of the mix of APC-conjugated antibodies. Post-incubation, beads were washed twice with 500 µL of MPB at $3000×g$ for 5 min. Each time of washing, the supernatant was aspirated, leaving a residual volume of 150 µL. The prepared beads were then analyzed by BD LSR Fortessa X-20 flow cytometer (BD Biosciences, San Jose, CA, USA). All mean fluorescence intensity was background-corrected according to the negative control.

For FCM comparison of sEV populations among plasma, SEC, SEC-DGUC, and dUC, $5×10^8$ of particles/sample were mixed with 15 µL of MACSPlex exosome capture beads and incubated at RT for 1 hr in the dark. The beads were then washed thrice with 500 µL of MPB at $3000×g$ for 5 min. 15 µL of the mix of APC-conjugated antibodies was added to each sample and incubated at RT for 1 hr in the dark. Post-incubation, beads were washed thrice with 500 µL of MPB at $3000×g$ for 5 min. Each time of washing, the supernatant was aspirated, leaving a residual volume of 150 µL. The prepared beads were then analyzed by BD LSR Fortessa X-20 flow cytometer as described above.

## sEV protein profiling by LC-MS/MS

Mass spectrometry experiments were carried out at the proteomics core facility in the School of Biological Sciences at Nanyang Technological University. For SEC-DGUC-1, which was obtained from 5 mL of plasma, the first step was to remove the iodixanol through dialysis using Pur-A-Lyzer Mini 6000 dialysis kit (PURN60030, Sigma-Aldrich, Co., STL, MO, USA). Then, SEC-DGUC-1 was subjected to in-solution digestion prior to fractionation on an Xbridge C18 column (4.6×250 mm, Waters, Milford, MA, USA) and subsequent analysis by LC-MS/MS. SEC-PF was subjected to in-solution digestion and subsequent analysis by LC-MS/MS.

For LC-MS/MS procedure, the peptides were separated and analyzed using a Dionex Ultimate 3000 RSLCnano system coupled to a Q Exactive instrument (Thermo Fisher Scientific, MA, USA). Separation was performed on a Dionex EASY-Spray 75 µm×10 cm column packed with PepMap C18 3 µm, 100 Å (Thermo Fisher Scientific) using solvent A (0.1% formic acid) and solvent B (0.1% formic acid in 100% ACN) at a flow rate of 300 nL/min with a 60 min gradient. Peptides were then analyzed on a Q Exactive apparatus with an EASY nanospray source (Thermo Fisher Scientific) at an electrospray potential of 1.5 kV.

Raw data files were processed and searched using Proteome Discoverer 2.1 (Thermo Fisher Scientific). The Mascot algorithm was then used for data searching to identify proteins using the following parameters: two missed cleavages; dynamic modifications were oxidation (+15.995 Da) (M) and phosphorylation (+79.966 Da) (S, T, Y). The static modification was carbamidomethyl (+57 Da) (C). Percolator was applied to filter out the false MS2 assignments at a strict false discovery rate of 1% and relaxed false discovery rate of 5%. Only proteins identified with two or more peptides were included in the final list. The protein database used for protein identification was Uniprot Human.

## Acknowledgements

This research has been funded through Nanyang Technological University Distinguished University Professorship awarded to Prof. Subra Suresh. MD acknowledges support through the Visiting Professorship from Nanyang Technological University and partial support from National Institutes of Health under Grant No. R01HL154150. We appreciate the guidance from Prof. Bernhard Otto Boehm on the

clinical aspects of sEV research. We thank Asst. Prof. Hou Han Wei for use of the Zetaview facility. We would like to express gratitude to Tan Tock Seng Hospital for providing the biobank samples.

## Additional information

### Funding

| Funder | Grant reference number | Author |
|---|---|---|
| Nanyang Technological University | | Fang Kong<br>Megha Upadya<br>Andrew SW Wong<br>Rinkoo Dalan<br>Ming Dao |
| National Institutes of Health | R01HL154150 | Ming Dao |

The funders had no role in study design, data collection and interpretation, or the decision to submit the work for publication.

### Author contributions

Fang Kong, Conceptualization, Data curation, Formal analysis, Validation, Investigation, Visualization, Methodology, Writing – original draft, Project administration, Writing – review and editing; Megha Upadya, Data curation, Formal analysis, Validation, Investigation, Writing – original draft, Writing – review and editing; Andrew SW Wong, Data curation, Validation, Investigation; Rinkoo Dalan, Conceptualization, Methodology, Project administration, Writing – review and editing; Ming Dao, Conceptualization, Resources, Supervision, Funding acquisition, Validation, Methodology, Writing – original draft, Writing – review and editing

### Author ORCIDs

Fang Kong https://orcid.org/0000-0002-1598-4335
Ming Dao https://orcid.org/0000-0001-5372-385X

### Ethics

All procedures involving peripheral blood specimens were approved by the Singapore National Health Group Domain Specific Review Board (the central ethics committee) and were mutually recognized by the Nanyang Technological University Institutional Review Board (IRB#2018/00671). All blood specimens were de-identified prior to their use in the experiments.

Reviewer #2 (Public review): https://doi.org/10.7554/eLife.92796.3.sa1
Author response https://doi.org/10.7554/eLife.92796.3.sa2

## Additional files

### Supplementary files

Supplementary file 1. LC-MS/MS proteins identified in SEC-DGUC-1 and SEC-particle fraction (PF) ranked according to emPAI values. (a) Liquid chromatography with tandem mass spectrometry (LC-MS/MS) proteins identified in size exclusion chromatography (SEC)-DGUC-1 by LC-MS/MS analysis ranked according to emPAI values. (b) LC-MS/MS proteins identified in SEC-particle fraction (PF) by LC-MS/MS analysis ranked according to emPAI values.

MDAR checklist

### Data availability

All data supporting the findings of this study are available within the article and its source data files. All numerical data underlying the figures are provided as source data files. Raw uncropped western blot images are provided for *Figures 6 and 7*. Experimental procedures are available on EV-TRACK (EV210379).

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
