## [Editor Report · eLife Assessment]

This work provides a simple, rapid and **valuable** protocol for the isolation of small extracellular vesicles from small volumes of plasma, using two well-known methodologies, in tandem: size exclusion chromatography (SEC) and density gradient ultracentrifugation (DGUC). The authors exhaustively test these methodologies separately and in combination, showing superior results for the SEC-DGUC in terms of purity and yield. The results obtained in this work are **convincing**, using multiple state-of-art methodologies for the characterization of the isolates that support their conclusions.

---

## [Referee Report · Reviewer #2 (Public review)]

Summary:

In this work, the authors manage to optimize a simple and rapid protocol using SEC followed by DGCU to isolate sEVs with adequate purity and yield from small volumes of plasma. Isolated fractions containing sEVs using SEC, DGCU, SEC-DGCU and DGCU-SEC are compared in terms of their yield, purity surface protein profile and RNA content. Although the combined use of these methodologies has already been evaluated in previous works, the authors manage to adapt them for the use of small volumes of plasma, which allows working in 1.5 mL tubes and reducing the centrifugation time to 2 hours.

The authors finally find that although both the SEC-DGCU and DGCU-SEC combinations achieve isolates with high purity, the SEC-DGCU combination results in higher yields.

This work provides an interesting tool for the rapid obtention of sEVs with sufficient yield and purity for detailed characterization which could be very useful in research and clinical therapy.

Strengths:

The work is well written and organized.

The authors clearly state the problem they want to address, that is, optimizing a method that allows sEV to be isolated from small volumes of plasma.

Although these methodologies have been tested in previous works, the authors manage to isolate sEVs of high purity and good performance through a simple and fast methodology.

The characteristics of all isolated fractions are exhaustively analyzed through various state-of-the-art methodologies.

They present a good interpretation of the results obtained through the methodologies used.

Weaknesses:

Although this work focuses on comparing different techniques and their combinations to find an optimal option, the authors could strengthen their analysis by using statistical methods that reliably show the differences between the explored techniques.

Comments on revisions:

Although superiority of the proposed method was demonstrated by other techniques, it is always advisable to calculate the differences between different methodologies through different statistical methods, whenever possible, to strengthen the obtained results.

---

## [Author Response]

The following is the authors’ response to the original reviews

**Public Reviews:**

**Reviewer #1 (Public Review):**
Summary:In their manuscript, Kong Fang et al describe a robust pipeline for the isolation of small extracellular vesicles through a combination of size exclusion chromatography and miniaturized density gradient separation. Subsequently, they prove that the method is reproducible and suitable for small-volume operations while at the same time not compromising the quality of vesicles.Strengths:The paper narrates a robust method for purifying high-quality sEVs from small amounts of blood plasma. They also demonstrate that through this approach, they can derive sEVs without compromising the protein composition, integrity of the vesicles, or contamination with other proteins or lipids.Weaknesses:The paper is a nice summary of how to enrich sEVs from blood samples. Although well performed and substantiated with data, the paper primarily deals with method development and optimisation.

We agree with the reviewer's assessment that this paper primarily focuses on the development and optimization of a method. Using this robust technique for isolating small extracellular vesicles (sEVs) from small blood volumes, our future research will investigate sEVs isolated from clinical samples, with a particular focus on their role in various diseases.

**Reviewer #2 (Public Review):**
Summary:In this work, the authors manage to optimize a simple and rapid protocol using SEC followed by DGCU to isolate sEVs with adequate purity and yield from small volumes of plasma. Isolated fractions containing sEVs using SEC, DGCU, SEC-DGCU, and DGCU-SEC are compared in terms of their yield, purity surface protein profile, and RNA content. Although the combined use of these methodologies has already been evaluated in previous works, the authors manage to adapt them for the use of small volumes of plasma, which allows working in 1.5 mL tubes and reducing the centrifugation time to 2 hours.The authors finally find that although both the SEC-DGCU and DGCU-SEC combinations achieve isolates with high purity, the SEC-DGCU combination results in higher yields.This work provides an interesting tool for the rapid obtention of sEVs with sufficient yield and purity for detailed characterization which could be very useful in research and clinical therapy.Strengths:- The work is well-written and organized.- The authors clearly state the problem they want to address, that is, optimizing a method that allows sEV to be isolated from small volumes of plasma.- Although these methodologies have been tested in previous works, the authors manage to isolate sEVs of high purity and good performance through a simple and fast methodology.- The characteristics of all isolated fractions are exhaustively analyzed through various state-of-the-art methodologies.- They present a good interpretation of the results obtained through the methodologies used.Weaknesses:- Lack of references that support some of the results obtained.- Although this work focuses on comparing different techniques and their combinations to find an optimal option, the authors do not use any statistical method that reliably shows the differences between these techniques, except when repeatability is measured.

We appreciate the reviewer's insightful comments and will incorporate the suggested missing references. We acknowledge that we did not perform statistical analyses when comparing the differences among the three methods. Nevertheless, the superiority of the SEC-DGUC method is evident from observations based on several independent characterization methods, including Cryo-EM, TEM, western blot, and total RNA quantification.

Firstly, repeated Cryo-EM observations consistently confirm that the SEC-alone method shows severe lipoprotein contamination while the SEC-DGUC method drastically reduces such lipoprotein contamination. In comparing the SEC-DGUC and DGUC-SEC methods, multiple independent characterization methods showed that the SEC-DGUC method yields significantly greater quantity of sEVs: (1) The western blot experiment showed much higher signal intensity for all four tested sEV markers (CD9, CD63, CD81, and TSG101), with estimated concentrations approximately 2.1, 2.1, 4.7, and 4.2 times higher than the DGUC-SEC method. (2) The total RNA analysis showed that SEC-DGUC-1 contained more than 4 times the total amount of RNA compared to DGUC-SEC-PF. (3) Establishing the normalization baseline, particle size distributions in SEC-DGUC-1 and DGUC-SEC-PF measured by TEM were found to be similar, suggesting comparable purity and distribution of the captured sEVs. For comparison purposes, within each independent characterization method, the same plasma source and total plasma volume were used, while across different methods, different plasma sources were used. These independent characterization methods have consistently demonstrated the superiority of the SEC-DGUC method over the DGUC-SEC or SEC-alone methods.

**Recommendations for the authors:**

**Reviewer #1 (Recommendations For The Authors):**
In my opinion, this work is elegantly designed and supported by data, which would motivate more studies related to blood-derived microvesicles in the context of infectious and systemic diseases. Overall, the manuscript is well-written and explained in sufficient detail. I only have minor comments.(1) Recruitment of volunteers for blood/plasma collection: there is a need for a statement that this was in accordance with ethical and biosafety regulations of the Institute/Clinic.

We added two sentences at the beginning of the Blood Collection section (under Materials and methods): “All procedures involving peripheral blood specimens were approved by the Singapore National Health Group Domain Specific Review Board (the central ethics committee) and were mutually recognized by the Nanyang Technological University Institutional Review Board (IRB#2018/00671). All blood specimens were de-identified prior to their use in the experiments.”

(2) Since this is a method development and validation article, it would be good to include an image of the iodixanol gradient with the high-density sEV zone, after centrifugation.

We have incorporated an image after centrifugation in Supplementary Figure 3.

(3) Although several sEV markers are shown in Figure 7A, flotillin is missing in this figure which was part of Figure 6B. Does flotillin show a different pattern? Flotillin is a DRM-associated marker, and hence may behave differently, would be interesting to add any insights.

We appreciate the reviewer’s careful observation. In Figure 6B, Flotillin was used to confirm the presence of sEVs in different density zones. However, for the purpose of comparing the yield between the SEC-DGUC and DGUC-SEC methods, as shown in Figure 7A, Flotillin was not included in the western blot analysis. No obvious pattern changes were observed in other sEV markers tested in both Figures 6B and 7A.

(4) Methods section of LC/MS analysis- which protein database was used for protein identification?

We added the following sentence at the end of the LC/MS analysis section: “The protein database used for protein identification was Uniprot Human.”

**Reviewer #2 (Recommendations For The Authors):**
In line 43 some references are needed.

We added this reference: EL Andaloussi, S., Mäger, I., Breakefield, X. et al. Extracellular vesicles: biology and emerging therapeutic opportunities. Nat Rev Drug Discov 12, 347–357 (2013). https://doi.org/10.1038/nrd3978

In line 107, please avoid using short forms such as "it's".

We have revised that to “it is.”

In line 153: "...separates low-density particles from those of high density, but a considerable amount of..." the word "but" should not be in the sentence.

We have removed “but” in this sentence.

In line 181 the authors establish that "Notably, SEC-PF exhibited a high level of ApoB and low expression of sEV markers." Is there any explanation for this?

SEC-PF represents the eluate from the SEC step, collected before the DGUC step. This fraction contains a mixture of lipoproteins and sEVs. Due to the overwhelming abundance of lipoproteins compared to sEVs, the western blot predictably shows a high level of ApoB with minimal expression of sEV markers. This highlights that SEC alone effectively reduces plasma protein content but does not efficiently remove lipoproteins. Figure 6C further illustrates this point, as cryo-EM images of SEC-PF reveal the presence of sEVs, which are vastly outnumbered by lipoproteins.

In line 198, the sentence "Theoretically, the DGUC-SEC protocol should also effectively isolate sEVs from plasma" need to be supported by references.See for instance:- Holcar M, Ferdin J, Sitar S, Tušek-Žnidarič M, Dolžan V, Plemenitaš A, Žagar E, Lenassi M. 2020. Enrichment of plasma extracellular vesicles for reliable quantification of their size and concentration for biomarker discovery. Sci Rep 10:21346. doi:10.1038/s41598-020-78422-y.- Jia Y, Yu L, Ma T, Xu W, Qian H, Sun Y, Shi H. 2022. Small extracellular vesicles isolation and separation: Current techniques, pending questions and clinical applications. Theranostics 12:6548-6575. doi:10.7150/thno.74305- Vergauwen G, Dhondt B, Van Deun J, De Smedt E, Berx G, Timmerman E, Gevaert K, Miinalainen I, Cocquyt V, Braems G, Van den Broecke R, Denys H, De Wever O, Hendrix A. 2017. Confounding factors of ultrafiltration and protein analysis in extracellular vesicle research. Sci Rep 7:2704. doi:10.1038/s41598-017-02599-y

We have added this reference: Holcar M, Ferdin J, Sitar S, Tušek-Žnidarič M, Dolžan V, Plemenitaš A, Žagar E, Lenassi M. 2020. Enrichment of plasma extracellular vesicles for reliable quantification of their size and concentration for biomarker discovery. Sci Rep 10:21346. https://doi.org/10.1038/s41598-020-78422-y.

In line 309 the authors establish that "NTA measured size distributions displayed well-overlapped histograms of particles". It is possible for the authors to analyze this overlapping using some statistical test as a chi-squared test?

We have conducted a statistical analysis of the histogram similarities using the Jensen-Shannon Divergence (JSD) method. This is reflected in the manuscript under the results section, “Repeatability and reliability of the SEC-DGUC protocol”, where we state: “We then compared size distributions for each plasma fraction using Jensen-Shannon Divergence (JSD). The JSD values, which are well below 0.1 (Figure 10B), indicate a consistent population of isolated particles, as further supported by Supplementary Figure 8.” Additionally, we included JSD values in the legend of Figure 10B: “JSD values for SEC-DGUC-1 to 4 are 0.015, 0.006, 0.001, and 0.002, indicating strong similarities among the histograms.” These additions demonstrate the robustness and repeatability of the SEC-DGUC protocol.

In line 360, "lasts ~ 16 hours or more." This statement needs a reference that supports this time.

We have added this reference: Vergauwen, G. et al. Robust sequential biophysical fractionation of blood plasma to study variations in the biomolecular landscape of systemically circulating extracellular vesicles across clinical conditions. J Extracell Vesicles 10, e12122 (2021).

In line 399, the reference format is different from the previously used format.

This is corrected. We thank the reviewer for this careful examination.

Line 466: This sentence is not quite clear. It can be understood that for every 0.5 mL of plasma, 2 mL of particle fraction are obtained and that for 6 mL of plasma, this method will give a total volume of 24 mL. However, it is not clear what is meant by the fact that it has been concentrated to 6 mL. While one can assume that those final 6 mL concentrates come from the initial 24 mL, perhaps the way this sentence was worded was not appropriate. I would recommend rewriting it for a simpler interpretation of how this method was performed.

We have changed the sentence to: “For the DGUC experiment using the 12 ml tube, 24 ml of PFs were obtained from 6 ml of plasma and subsequently concentrated to 6 ml. The 6 ml of concentrated PFs were then transferred to a Beckman Coulter ultra-clear centrifuge tube (344059, Beckman Coulter, USA) for further processing.”

Line 519: The authors established a second dilution to avoid absorbance values above 1.2. Is there any justification for this value, taking into account that the Lambert-Beer law presents more precision in the absorbance range of 0.2 to 0.8?

We have added this reference: here

Line 519-520: "Also included were water and 0.25 M sucrose as blanks". Perhaps authors could consider rephrasing this sentence.

We have changed the sentence to: “The absorbance measurements were made against water and 0.25 M sucrose blanks.”

In line 520, the sentence must say "each sample was made by triplicate".

We have changed the sentence to: “Each sample was prepared by triplicate to reduce error.” We thank the reviewer for this suggestion.

Line 673: The phrase "0.1% formic acid in 100% ACN" would be better, in my opinion, if it said "0.1% formic acid in ACN".

Yes, these two expressions have the same meaning. However, to ensure clarity, we have updated the description to “0.1% formic acid in ACN.”. We thank reviewer for this suggestion.

Supplementary Figure 1: in the Figure caption there is an error in the numbering: at the end, where it is written (E), it should be (F). Please, correct this.

We have made the necessary correction and sincerely appreciate the reviewer’s attentiveness.

Supplementary Figure 5: Some sEVs are hard to visualize due to poor image resolution. Is there any possibility for the authors to enhance these images?

We thank the reviewer for this valuable comment. To improve the visual clarity of the images, we have opted to display four sub-figures instead of nine.